# Gyrogroup Batch Normalization

**Ziheng Chen**[1]*, **Yue Song**[1], **Xiao-Jun Wu**[2] **& Nicu Sebe**[1]
[1] University of Trento, [2] Jiangnan University

## Abstract

Several Riemannian manifolds in machine learning, such as Symmetric Positive Definite (SPD), Grassmann, spherical, and hyperbolic manifolds, have been proven to admit gyro structures, thus enabling a principled and effective extension of Euclidean Deep Neural Networks (DNNs) to manifolds. Inspired by this, this study introduces a general Riemannian Batch Normalization (RBN) framework on gyrogroups, termed GyroBN. We identify the least requirements to guarantee GyroBN with theoretical control over sample statistics, referred to as *pseudo-reduction* and *gyroisometric gyrations*, which are satisfied by all the existing gyrogroups in machine learning. Besides, our GyroBN incorporates several existing normalization methods, including the one on general Lie groups and different types of RBN on the non-group SPD geometry. Lastly, we instantiate our GyroBN on the Grassmannian and hyperbolic spaces. Experiments on the Grassmannian and hyperbolic networks demonstrate the effectiveness of our GyroBN. The code is available at https://github.com/GitZH-Chen/GyroBN.git.

## 1 Introduction

Deep Neural Networks (DNNs) on Riemannian manifolds have gained increasing interest in various machine learning applications, such as computer vision (Huang et al., 2017; Huang & Van Gool, 2017; Huang et al., 2018; Skopek et al., 2019; Wang et al., 2022b;a; Chen et al., 2023c; Gao et al., 2023; Wang et al., 2024b; Chen et al., 2025), natural language processing (Ganea et al., 2018; Shimizu et al., 2020), drone classification (Brooks et al., 2019; Chen et al., 2024a), human neuroimaging (Pan et al., 2022; Kobler et al., 2022a; Ju et al., 2024; Wang et al., 2024a), medical imaging (Huang et al., 2019; Chakraborty et al., 2020), node and graph classification (Chami et al., 2019; Dai et al., 2021; Zhao et al., 2023; Chen et al., 2023b; Nguyen et al., 2024; Chen et al., 2024c). As core techniques in DNNs, normalization techniques (Ioffe & Szegedy, 2015; Ba et al., 2016; Ulyanov et al., 2016; Wu & He, 2018; Chen et al., 2023a) have also been extended into different geometries.

However, most existing Riemannian normalization methods are designed for a selected few geometries or fail to normalize the sample statistics. Brooks et al. (2019); Kobler et al. (2022b;a) introduced Riemannian Batch Normalization (RBN) on SPD manifolds under the specific Affine-Invariant Metric (AIM). Chakraborty (2020) generalized this idea and proposed a Riemannian normalization framework over homogeneous spaces. However, this approach cannot generally normalize the sample statistics. Similar formulation and issue can also be found in Lou et al. (2020, Alg. 2) and Bdeir et al. (2024, Sec. 4.2). Besides, Chakraborty (2020) also developed a Riemannian normalization for matrix Lie groups. Although this approach can control first- and second-order statistics, it is limited to a specific type of distance (Chakraborty, 2020, Sec. 3.2). Chen et al. (2024b) took one step further and developed RBN over the general Lie group, referred to as LieBN. Although LieBN can effectively normalize sample statistics, many geometries do not admit a group structure. In summary, the existing RBN methods fail to normalize manifold-valued samples in a principled manner.

Recently, building Riemannian networks based on gyro structures has shown notable success across various geometries, including Symmetric Positive Definite (SPD) (Nguyen, 2022a;b), Grassmannian (Nguyen, 2022b), hyperbolic (Ganea et al., 2018), and spherical (Skopek et al., 2019) manifolds. Gyro structures, natural extensions of vector structures, offer powerful mathematical tools for building Riemannian neural networks. Moreover, gyrogroups naturally encompass Lie groups and extend to non-group geometries. For instance, AIM on the SPD manifold, as well as Grassmannian, hyperbolic, and spherical manifolds, do not form Lie groups but instead gyrogroups.

---

*Correspondence to ziheng_ch@163.com

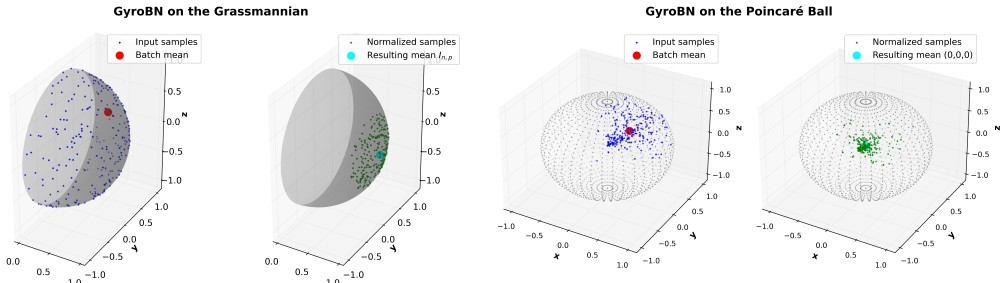

Figure 1: Illustration of GyroBN on the Grassmannian and hyperbolic spaces. As $\mathrm{Gr}(1,3)$ is homeomorphic to the real projective space $\mathbb{RP}^2$, we illustrate $\mathrm{Gr}(1,3)$ as the unit hemisphere with antipodal points identified. We set the bias as $I_{1,3} = (1,0,0)^\top$ and the scaling scalar as $0.2$ for better illustration. For the hyperbolic space, we visualize the GyroBN on the Poincaré ball model $\mathbb{P}^3_{-1}$, which is the interior of the unit sphere in $\mathbb{R}^3$. We set bias and shift as zero vector and $0.7$, respectively.

Table 1: Comparison of previous RBN methods with our GyroBN, where M and V denote the sample mean and variance. Compared with the existing RBN methods, our GyroBN can normalize sample statistics in a principled manner. Besides, several previous RBN methods with theoretical control over sample statistics are special cases of our GyroBN.

| Method | Controllable Statistics | Applied Geometries | Incorporated by GyroBN |
|---|---|---|---|
| SPDBN (Brooks et al., 2019) | M | SPD manifolds under AIM | ✓ |
| SPDBN (Kobler et al., 2022b) | M+V | SPD manifolds under AIM | ✓ |
| SPDDSMBN (Kobler et al., 2022a) | M+V | SPD manifolds under AIM | ✓ |
| ManifoldNorm (Chakraborty, 2020, Algs. 1-2) | N/A | Riemannian homogeneous space | ✗ |
| ManifoldNorm (Chakraborty, 2020, Algs. 3-4) | M+V | Matrix Lie groups under the distance $d(X,Y) = \left\lVert \mathrm{mlog}\left(X^{-1}Y\right) \right\rVert$ | ✓ |
| RBN (Lou et al., 2020, Alg. 2) | N/A | Geodesically complete manifolds | ✗ |
| LieBN (Chen et al., 2024b) | M+V | General Lie groups | ✓ |
| GyroBN | M+V | Pseudo-reductive gyrogroups with gyro isometric gyrations | N/A |

Based on the above analysis, this paper introduces GyroBN, an RBN framework for general gyrogroups. We use the gyro addition, subtraction, and scalar product to extend the centering (vector subtraction), biasing (vector addition), and scaling (vector scalar product) in the Euclidean BN into manifolds in a principled manner. For broader applicability and in-depth theoretical analysis, we adapt the existing gyrogroup into a more relaxed structure, termed *pseudo-reductive gyrogroup*. Our theoretical analysis shows that pseudo-reductive gyrogroups with gyroisometric gyrations can enable GyroBN with theoretical control over sample statistics. More importantly, these requirements are satisfied by all the existing gyrogroups in machine learning. Therefore, compared with the existing RBN methods, our GyroBN can normalize sample statistics in a principled manner. Besides, several existing RBN methods are incorporated by our GyroBN as special cases, including the LieBN on Lie groups, such as three SPD Lie groups and rotation matrices, and different types of RBN on the non-group SPD geometry. We provide a detailed comparison in Tab. 1. Empirically, we instantiate our GyroBN on the Grassmannian and hyperbolic spaces, as illustrated in Fig. 1. To the best of our knowledge, our Grassmannian GyroBN is the first implementation of Grassmannian RBN. Experiments on the Grassmannian and hyperbolic networks validate the effectiveness of our GyroBN.

Our **main theoretical contributions** are summarized as follows: (1) We propose the pseudo-reductive gyrogroup, a relaxed structure of the gyrogroup, and present relevant theoretical analyses; (2) We identify the requirements that guarantee our GyroBN with theoretical control over sample statistics, *i.e.,* pseudo-reduction and gyroisometric gyrations; (3) We propose a GyroBN framework for RBN over general gyrogroups, which can be manifested in various geometries in a plug-and-player manner. (4) We implement our GyroBN on the Grassmannian and hyperbolic spaces. Extensive experiments on popular Grassmannian and hyperbolic networks validate the effectiveness of our framework.

**Main theoretical results:** Def. 3.1 relaxes the existing gyrogroup into the pseudo-reductive gyrogroup. Prop. 3.2 reveals that the non-reductive Grassmannian gyrogroup is, in fact, pseudo-reductive. Thms. 3.3 and 3.5 highlight that the invariance of gyronorm under gyrations is crucial for enabling several operators to act as gyroisometries. Prop. 3.6 confirms that all gyrogroups listed in Tab. 2 are pseudo-reductive and their gyrations are gyroisometries. These two properties are essential for enabling GyroBN in Alg. 1 to normalize sample statistics, which are formalized in Thm. 4.1. Sec. 5 discusses how several existing RBN methods are special cases of our GyroBN. Lastly, Sec. 6 manifest our GyroBN on the Grassmannian and hyperbolic spaces, where Prop. 6.1 discuss the efficient implementation on the Grassmannian. Due to page limits, all the proofs are presented in App. G.

## 2 PRELIMINARIES

This section recaps gyrogroups (Ungar, 2009) and several concrete gyrogroups in machine learning.

**Definition 2.1** (Gyrogroups (Ungar, 2009)). Given a nonempty set $G$ with a binary operation $\oplus : G \times G \to G$, $\{G, \oplus\}$ forms a gyrogroup if its binary operation satisfies the following axioms for any $a, b, c \in G$:

(G1) There is at least one element $e \in G$ called a left identity (or neutral element) such that $e \oplus a = a$.

(G2) There is an element $\ominus a \in G$ called a left inverse of $a$ such that $\ominus a \oplus a = e$.

(G3) There is an automorphism $\text{gyr}[a, b] : G \to G$ for each $a, b \in G$ such that

$$a \oplus (b \oplus c) = (a \oplus b) \oplus \text{gyr}[a, b]c \quad \text{(Left Gyroassociative Law).} \quad (1)$$

The automorphism $\text{gyr}[a, b]$ is called the gyroautomorphism, or the gyration of $G$ generated by $a, b$.
(G4) Left reduction law: $\text{gyr}[a, b] = \text{gyr}[a \oplus b, b]$.

**Definition 2.2** (Gyrocommutative Gyrogroups (Ungar, 2009)). A gyrogroup $\{G, \oplus\}$ is gyrocommutative if it satisfies

$$a \oplus b = \text{gyr}[a, b](b \oplus a) \quad \text{(Gyrocommutative Law).} \quad (2)$$

**Definition 2.3** (Nonreductive Gyrogroups (Nguyen, 2022a)). A groupoid $\{G, \oplus\}$ is a nonreductive gyrogroup if it satisfies axioms (G1), (G2), and (G3).

Intuitively, gyrogroups are natural generalizations of groups. Unlike groups, gyrogroups are non-associative but have gyroassociativity characterized by gyrations. Since all gyrations in any (Lie) group are the identity map, every (Lie) group is automatically a gyrogroup.

As shown by Nguyen & Yang (2023), given $P$, $Q$ and $R$ in a manifold $\mathcal{M}$ and $t \in \mathbb{R}$, the gyro structures can be defined as:

$$\text{Gyro addition: } P \oplus Q = \text{Exp}_P \left( \text{PT}_{E \to P} \left( \text{Log}_E(Q) \right) \right), \quad (3)$$

$$\text{Gyro scalar product: } t \odot P = \text{Exp}_E \left( t \, \text{Log}_E(P) \right), \quad (4)$$

$$\text{Gyro inverse: } \ominus P = -1 \odot P = \text{Exp}_E \left( -\text{Log}_E(P) \right), \quad (5)$$

$$\text{Gyration: } \text{gyr}[P, Q]R = (\ominus(P \oplus Q)) \oplus (P \oplus (Q \oplus R)), \quad (6)$$

$$\text{Gyro inner product: } \langle P, Q \rangle_{\text{gr}} = \langle \text{Log}_E(P), \text{Log}_E(Q) \rangle_E, \quad (7)$$

$$\text{Gyro norm: } \|P\|_{\text{gr}} = \langle P, P \rangle_{\text{gr}}, \quad (8)$$

$$\text{Gyrodistance: } \text{d}_{\text{gry}}(P, Q) = \|\ominus P \oplus Q\|_{\text{gr}}, \quad (9)$$

where $E$ is the gyro identity element, and $\text{Log}_E$ and $\langle \cdot, \cdot \rangle_E$ is the Riemannian logarithm and metric at $E$. A bijection $\omega : G \to G$ is called gyroisometry, if it preserves the gyrodistance

$$\text{d}_{\text{gry}}(\omega(P), \omega(Q)) = \text{d}_{\text{gry}}(P, Q). \quad (10)$$

Note that the gyro scalar product $\odot$ is required for a gyrogroup to form a gyrovector space (Nguyen, 2022b). Although this paper only involves gyrogroups, we also recap gyrovector spaces in App. C.

Several geometries in machine learning admit a gyro structure defined in Eq. (3)-Eq. (9) and form a (nonreductive) gyrogroup, such as Affine-Invariant Metric (AIM) (Pennec et al., 2006), Log-Euclidean Metric (LEM) (Arsigny et al., 2005), and Log-Cholesky Metric (LCM) (Lin, 2019) on the SPD manifold $\mathcal{S}_{++}^n$ (Nguyen, 2022a), Orthonormal Basis (ONB) perspective $\text{Gr}(p, n)$ (Bendokat et al., 2024) and projector perspective $\widetilde{\text{Gr}}(p, n)$ (Bendokat et al., 2024) for the Grassmannian (Nguyen,

2022a; Nguyen & Yang, 2023), Poincaré ball $\mathbb{P}_K^n$ for the hyperboloid (Ungar, 2009; Ganea et al., 2018), and projected hypersphere $\mathbb{D}_K^n$ for the hypersphere (Skopek et al., 2019). Besides, the gyrogroups proposed by Nguyen (2022b) on the SPD manifold under the LEM and LCM coincide with the Lie groups proposed by Arsigny et al. (2005); Lin (2019).

We denote $\mathcal{M}_K$ as $\mathbb{P}_K^n(K < 0)$, $\mathbb{D}_K^n(K > 0)$ and $\mathbb{R}^n(K = 0)$, respectively. $\mathcal{M}_K$ is known as the Constant Curvature Space (CCS) (Do Carmo & Flaherty Francis, 1992). We summarize all the necessary gyro properties in Tab. 2.

Table 2: Gyrogroup properties on several geometries. Related notations are defined in App. C.3.2.

| Geometry | Symbol | $P \oplus Q$ or $x \oplus y$ | $E$ | $\ominus P$ or $\ominus x$ | Lie group | Gyrogroup | References |
|---|---|---|---|---|---|---|---|
| AIM $\mathcal{S}_{++}^n$ | $\oplus^{\mathrm{AI}}$ | $P^{\frac{1}{2}}QP^{\frac{1}{2}}$ | $I_n$ | $P^{-1}$ | ✗ | ✓ | (Nguyen, 2022b) |
| LEM $\mathcal{S}_{++}^n$ | $\oplus^{\mathrm{LE}}$ | $\mathrm{mexp}(\mathrm{mlog}(P) + \mathrm{mlog}(Q))$ | $I_n$ | $P^{-1}$ | ✓ | ✓ | (Arsigny et al., 2005) (Nguyen, 2022b) (Lin, 2019) |
| LCM $\mathcal{S}_{++}^n$ | $\oplus^{\mathrm{LC}}$ | $\psi_{\mathrm{LC}}^{-1}(\psi_{\mathrm{LC}}(P) + \psi_{\mathrm{LC}}(Q))$ | $I_n$ | $\psi_{\mathrm{LC}}(-\psi_{\mathrm{LC}}(P))$ | ✓ | ✓ | (Nguyen, 2022b) (Chen et al., 2024e) |
| $\widetilde{\mathrm{Gr}}(p,n)$ | $\widetilde{\oplus}^{\mathrm{Gr}}$ | $\mathrm{mexp}(\Omega)Q\,\mathrm{mexp}(-\Omega)$ | $\widetilde{I}_{p,n}$ | $\mathrm{mexp}(-\Omega)\widetilde{I}_{p,n}\,\mathrm{mexp}(\Omega)$ | ✗ | Non-reductive | (Nguyen, 2022a) |
| $\mathrm{Gr}(p,n)$ | $\oplus^{\mathrm{Gr}}$ | $\mathrm{mexp}(\Omega)V$ | $I_{p,n}$ | $\mathrm{mexp}(-\Omega)I_{p,n}$ | | | (Nguyen & Yang, 2023) |
| $\mathcal{M}_K$ | $\oplus_K$ | $\frac{\left(1-2K\langle x,y\rangle-K\|y\|^2\right)x+\left(1+K\|x\|^2\right)y}{1-2K\langle x,y\rangle+K^2\|x\|^2\|y\|^2}$ | $0$ | $-x$ | ✗(✓for $K=0$) | ✓ | (Ungar, 2009) (Ganea et al., 2018) (Skopek et al., 2019) |

## 3  PSEUDO-REDUCTIVE GYROGROUPS

As shown in Tab. 2, the Grassmannian does not satisfy left reduction (G4) in Def. 2.1. However, we find that a relaxed version of (G4) is necessary to guarantee the sample normalization over gyrogroups. Therefore, this section proposes an intermediate between the gyrogroup and the nonreductive one, called pseudo-reductive gyrogroups.

Unless specifically emphasized, the gyro structure in this paper is defined as Eq. (3)-Eq. (9). Given a gyrogroup $\{G, \oplus\}$, the left gyrotranslation by $P \in G$ is defined as

$$L_P : G \to G, \text{ with } L_P(Q) = P \oplus Q, \forall Q \in G. \tag{11}$$

If any left gyrotranslation is a gyroisometry, we can use gyrotranslation to center samples. Nguyen & Yang (2023) show that any left gyrotranslation on the SPD and Grassmannian manifolds is a gyroisometry. However, the proof relies on the left cancellation law of gyrogroups, which does not hold for nonreductive gyrogroups. Therefore, the proof is questionable for the Grassmannian. This subsection proposes an intermediate structure, referred to as pseudo-reductive gyrogroups, which can support left cancellation law in general and, therefore, gyroisometry of left gyrotranslation. We illustrate the derivation logic in Fig. 2.

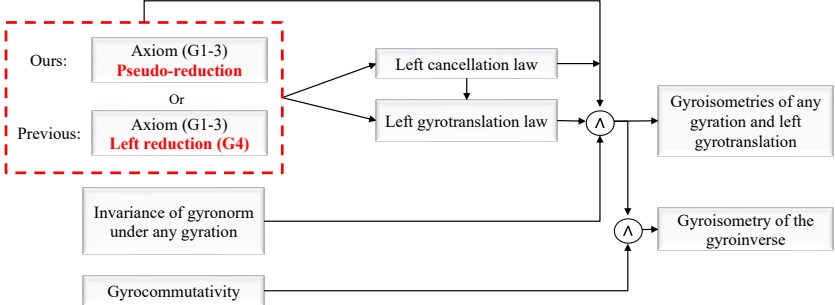

Figure 2: The conceptual comparison of derivation logic of gyroisometries of our work against previous work (Nguyen & Yang, 2023), where the left gyrotranslation law is presented in Lem. G.1. The previous work proves the results on the SPD and Grassmannian manifolds in a case-by-case manner. In contrast, we relax the left reduction into pseudo-reduction and give a general theoretical framework. Our framework also corrects the proof for the Grassmannian cases.

**Definition 3.1** (Pseudo-reductive Gyrogroups). A groupoid $\{G, \oplus\}$ is a pseudo-reductive gyrogroup if it satisfies axioms (G1), (G2), (G3) and the following pseudo-reductive law:

$$\mathrm{gyr}[X, P] = \mathbb{1}, \text{ for any left inverse } X \text{ of } P \text{ in } G, \tag{12}$$

where $\mathbb{1}$ is the identity map.

Eq. (12) can be intuitively viewed as the intermediate between reduction and non-reduction. For gyrogroups, Eq. (12) can be directly obtained from the left gyroassociativity (G3) and reduction (G4) (Ungar, 2009, p. 12). However, there is no theoretical guarantee that Eq. (12) holds for general non-reductive gyrogroups. Therefore, we name Eq. (12) as pseudo-reduction. Nevertheless, for the specific non-reductive Grassmannian, it is indeed pseudo-reductive.

**Proposition 3.2.** [↓] $\mathrm{Gr}(p, n)$ *and* $\widetilde{\mathrm{Gr}}(p, n)$ *form pseudo-reductive and gyrocommutative gyrogroups.*

Our pseudo-reductive gyrogroup naturally generalizes the vanilla gyrogroup, as it shares most of the basic properties of gyrogroups (Ungar, 2009, Thms. 1.13 - 1. 14), which are detiled in Thm. D.1. The most related property in Thm. D.1 is the left cancellation law, one of the key prerequisites for gyro translation as gyroisometry. Note that the left cancellation on the gyrogroup comes from left gyro associativity and Eq. (12) (Ungar, 2009, p. 12). Therefore, left cancellation does not generally hold for non-reductive gyrogroups, but exists in pseudo-reductive gyrogroups. Next, we point out an iff statement about gyroisometry, which will be useful in the following.

**Theorem 3.3.** [↓] *Given a pseudo-reductive gyrogroup* $\{G, \oplus\}$*,* $\mathrm{gyr}[P, Q]$ *preserves gyronorm for any* $P, Q \in G$*, iff* $\mathrm{gyr}[P, Q]$ *is a gyroisometry for any* $P, Q \in G$*.*

We find that the isometry of any gyration is a key prerequisite for other gyro operators as isometries.
**Definition 3.4** (Gyro Left-invariance). The gyrodistance or gyrogroup is gyro left-invariant if any left gyrotranslation is a gyroisometry.
**Theorem 3.5** (Gyroisometries). [↓] *Given a pseudo-reductive gyrogroup* $\{G, \oplus\}$ *with any* $\mathrm{gyr}[\cdot, \cdot]$ *as a gyroisometry, then we have the following:*

1. *The gyrodistance (Eq. (9)) is gyro left-invariant;*

2. *Symmetry of the gyrodistance:* $\forall P, Q \in G, \mathrm{d_{gry}}(P, Q) = \mathrm{d_{gry}}(Q, P)$*;*

3. *If* $\{G, \oplus\}$ *is gyrocommutative, then the gyroinverse is a gyroisometry;*

**Proposition 3.6.** [↓] *For every (pseudo-reductive) gyrogroup in Tab. 2, the gyrodistance is identical to the geodesic distance (therefore symmetric). The gyroinverse, any gyration and any left gyrotranslation are gyroisometries.*

*Credit of the proof.* For the Grassmannian and SPD manifolds, the isometries of gyration, gyroinverse, and left gyrotranslation have been proven by Nguyen & Yang (2023, Thms. 2. 12 - 2. 14 and 2.16 - 2. 18). Nevertheless, in the proof of Thms. 2. 12 - 2. 14, the authors view the non-reductive Grassmannian as a gyrogroup, as they use the left cancellation property of the gyrogroup. Fortunately, our Prop. 3.2 and Thm. D.1 shows that the Grassmannian still enjoys left cancellation. Therefore, all the results on isometries in their Thms. 2.12-2.14 are correct. Nevertheless, these results on the SPD and Grassmannian manifolds can be directly obtained by our Thms. 3.3 and 3.5, as these gyrogroups are pseudo-reductive and the gyrations preserve the gyronorm. Besides, we show the expressions for the gyrodistance of all gyrogroups in Tab. 2, and the associated gyroisometries on $\mathcal{M}_K$, both of which is none-trivial. The detailed proof is presented in App. G.4. □

## 4 GYROBN ON GENERAL PSEUDO-REDUCTIVE GYROGROUPS

Prop. 3.6 shows that several geometries enjoy isometric gyrotranslation, offering a theoretical foundation for normalizing samples over gyrogroups in a principled manner. Inspired by this, this section develops Riemannian Batch Normalization (RBN) for general pseudo-reductive gyrogroups, referred to as GyroBN. In the following, $\{\mathcal{M}, \oplus\}$ is assumed as a pseudo-reductive gyrogroup with the gyro structure defined as Eq. (3)-Eq. (9)[1].

### 4.1 EUCLIDEAN BATCH NORMALIZATION REVISITED

As the core operations of different Euclidean normalization variants (Ioffe & Szegedy, 2015; Ba et al., 2016; Ulyanov et al., 2016; Wu & He, 2018) are similar, this paper focuses on BN. Given a batch of activations $\{x_{i \ldots N}\}$, the core operations of BN can be expressed as:

$$\forall i \leq N, x_i \leftarrow \gamma \frac{x_i - \mu}{\sqrt{v^2 + \epsilon}} + \beta \tag{13}$$

---

[1]In our GyroBN, $\odot$ is not required to comply with the axioms of the gyrovector space (Def. C.4).

where $\mu$, $v^2$, $\gamma$, and $\beta$ are the sample mean, sample variance, scaling parameter, and biasing parameter, respectively.

## 4.2 GYROBN

To generalize the Euclidean BN into gyrogroups, we first define sample mean, sample variance, centering, biasing, and scaling over gyrogroups. Then, we introduce our GyroBN framework with a theoretical analysis of the ability to normalize sample statistics.

We define the gyromean as the Fréchet mean under the gyrodistance:

$$M = \text{FM}(\{P_i\}) = \underset{Q \in \mathcal{M}}{\arg\min} \frac{1}{N} \sum_{i=1}^{N} \text{d}^2_{\text{gry}}(P_i, Q) \tag{14}$$

The gyrovariance is the Fréchet variance, *i.e.,*the minimization of the rightest hand side of Eq. (14). When the distance in Eq. (14) is the geodesic distance, the Fréchet mean and variance are known as the Riemannian mean and variance. Although the gyromean and gyrovariance are not necessarily the same as the Riemannian ones, Prop. 3.6 indicates the equivalence for the gyrogroups in Tab. 2.

Easy computation shows that the centering and biasing in the Euclidean BN (Eq. (13)) can be viewed as gyro addition (Eq. (3)) in $\mathbb{R}^n$, and the scaling can be viewed as gyro scalar product (Eq. (4)), scaling in the tangent space at the identity element. Inspired by this, we define the normalization over gyrogroups by gyro addition and gyro scalar product. Given a batch of activations $\{P_{i...N} \in \mathcal{M}\}$, we defined the core operations of GyroBN as

$$\forall i \leq N, \tilde{P}_i = \overbrace{B \oplus}^{\text{Biasing}} \left( \overbrace{\frac{s}{\sqrt{v^2 + \epsilon}} \odot}^{\text{Scaling}} \left( \overbrace{\ominus M \oplus P_i}^{\text{Centering}} \right) \right), \tag{15}$$

where $M \in \mathcal{M}$ and $v^2$ are gyromean and gyrovariance, $B \in \mathcal{M}$ is the biasing parameter, $s \in \mathbb{R}$ is the scaling parameter, and $\epsilon$ is a small value for numerical stability.

**Theorem 4.1** (Homogeneity). [↓] *Supposing $\{\mathcal{M}, \oplus\}$ is a pseudo-reductive gyrogroup with any gyration $\text{gyr}[\cdot, \cdot]$ as a gyroisometry, for $N$ samples $\{P_{i...N} \in \mathcal{M}\}$, we have the following properties:*

$$\textit{Homogeneity of gyromean: } \text{FM}(\{B \oplus P_i\}) = B \oplus \text{FM}(\{P_i\}), \forall B \in \mathcal{M}, \tag{16}$$

$$\textit{Homogeneity of dispersion from E: } \frac{1}{N} \sum_{i=1}^{N} \text{d}^2_{\text{gry}}(t \odot P_i, E) = \frac{t^2}{N} \sum_{i=1}^{N} \text{d}^2_{\text{gry}}(P_i, E), \tag{17}$$

The most important property of the Euclidean BN (Eq. (13)) lies in its ability to normalize data distribution by the control over sample mean and variance. Similarly, our formulation in Eq. (15) can also normalize gyromean and gyrovariance. Specifically, given a pseudo-reductive gyrogroup with isometric gyrations, Eq. (16) indicates that the centering and biasing can transfer the gyromean, while Eq. (17) can scale the sample variance, since after centering, the resulting gyromean is the identity element $E$.

To finalize our GyroBN, we define the running mean updates over gyrogroups as the binary barycenter based on gyrodistance:

$$\text{Bar}_\eta(P_1, P_2) = \arg\min_{Q \in \mathcal{M}} \left( \eta \text{d}^2_{\text{gry}}(P_1, Q) + (1 - \eta) \text{d}^2_{\text{gry}}(P_2, Q) \right), \text{ with } \eta \in [0, 1]. \tag{18}$$

Notably, when the gyrodistance is identical to the geodesic distance, the binary barycenter can be calculated by geodesic.

With all the above ingredients, the general framework for our GyroBN is presented in Alg. 1. Thm. 4.1 indicates that given a pseudo-reductive gyrogroup with any gyration as a gyroisometry, our GyroBN enjoys a theoretical guarantee of control over the gyro statistics. Specifically, for the gyrogroups in Tab. 2, our GyroBN can control the gyromean and gyrovariance. Besides, as the gyromean and gyrovariance are identical to the Riemannian counterparts, the GyroBNs on these gyrogroups also normalize the Riemannian statistics. Especially, simple computation shows that our GyroBN recovers the standard Euclidean BN (Ioffe & Szegedy, 2015) when $\mathcal{M} = \mathbb{R}^n$.

---

**Algorithm 1:** Gyrogroup Batch Normalization (GyroBN)

| | |
|---|---|
| **Require** | : batch of activations $\{P_{1\ldots N} \in \mathcal{M}\}$, small positive constant $\epsilon$, and momentum $\eta \in [0, 1]$, running mean $M_r$, running variance $v_r^2$, biasing parameter $B \in \mathcal{M}$, scaling parameter $s \in \mathbb{R}$. |
| **Return** | : normalized batch $\{\tilde{P}_{1\ldots N} \in \mathcal{M}\}$ |

1 **if** *training* **then**
2     Compute batch mean $M_b$ and variance $v_b^2$ of $\{P_{1\ldots N}\}$;
3     Update running statistics $M_r = \mathrm{Bar}_\gamma(M_b, M_r), v_r^2 = \gamma v_b^2 + (1 - \gamma)v_r^2$;
4 **end**
5 $(M, v^2) = (M_b, v_b^2)$ **if** *training* **else** $(M_r, v_r^2)$
6 $\forall i \leq N, \tilde{P}_i = B \oplus \left( \frac{s}{\sqrt{v^2 + \epsilon}} \odot (\ominus M \oplus P_i) \right)$

---

## 5 A GYRO PERSPECTIVE FOR THE EXISTING RIEMANNIAN NORMALIZATIONS

Several existing Riemannian normalization methods on different geometries enjoy theoretical control of sample mean and variance, including LieBN on general Lie groups (Chen et al., 2024b), and SPDBNs based on the specific AIM geometry (Brooks et al., 2019; Kobler et al., 2022b;a). This subsection further reveals that they are concrete implementations of our GyroBN.

### 5.1 LIEBN AS A SPECIAL CASE OF GYROBN

Chakraborty (2020, Algs. 3-4) first proposed Riemannian normalization on matrix Lie groups under a specific distance. Chen et al. (2024b) extended their framework into general Lie groups, referred to as LieBN, with a theoretical control over Riemannian mean and variance. This subsection shows that LieBN is indeed a special case of our GyroBN.

LieBN is established under a left-invariant metric on the Lie group. The centering and biasing are defined by left group translation, and scaling is defined by the scaling on the tangent space at the identity element (Chen et al., 2024b, Eq.13-15). As every Lie group is automatically a gyrogroup, the gyrotranslation is the exact group translation. Therefore, the centering, biasing, and scaling are the same under GyroBN and LieBN. However, the mean, variance, and running mean updates on LieBN are defined based on the geodesic distance. In contrast, the counterparts on the GyroBN are based on gyrodistance. Nevertheless, the following proposition demonstrates the equivalence of these operators under the GyroBN with LieBN.

**Proposition 5.1.** [↓] *Given a Lie group with a left-invariant metric, the gyrodistance and geodesic distance are identical. The GyroBN is, therefore, identical to the LieBN (Chen et al., 2024b, Alg. 1).*

Chen et al. (2024b) implemented LieBN on four left-invariant geometries, including SPD manifold with AIM[2], LEM and LCM, and rotation matrices. According to Prop. 5.1, these implementations are immediately the special cases of our GyroBN.

### 5.2 AIM-BASED SPDBNS AS SPECIAL CASES OF GYROBN

Several RBNs on the SPD manifold were developed based on AIM (Brooks et al., 2019; Kobler et al., 2022b;a). The core operations of these approaches can be expressed as the following:

$$\text{Normalization: } \forall i \leq N, \tilde{P}_i = B^{\frac{1}{2}} \left( M^{-\frac{1}{2}} P_i M^{-\frac{1}{2}} \right)^{\frac{s}{\sqrt{v^2 + \epsilon}}} B^{\frac{1}{2}} \tag{19}$$

where $M$ and $v^2$ are the Riemannian mean and variance, *i.e.,*the Fréchet mean and variance under the geodesic distance. The running mean is updated by binary barycenter under the geodesic distance.

Prop. 3.6 indicates that, under the AIM geometry, the gyrodistance is identical to the geodesic distance. Therefore, the gyromean and gyrovariance are identical to the Riemannian mean and variance. The running mean updates are also identical under the gyrodistance and geodesic distance. Besides, simple computations show that Eq. (19) is exactly the specific implementation of Eq. (15) under $\{\mathcal{S}_{++}^n, \oplus^{\mathrm{AI}}, g^{\mathrm{AI}}\}$, where $g^{\mathrm{AI}}$ denotes AIM. Therefore, the SPDBNs developed by Brooks et al. (2019); Kobler et al. (2022b;a) are also special cases of our GryoBN.

---

[2]AIM is left-invariant w.r.t. the Lie group operation $P \oplus^{\mathrm{LieAI}} Q = LQL^\top$ with $L$ as the Cholesky factor of $P = LL^\top$ (Thanwerdas & Pennec, 2022). This group structure differs from the one presented in Tab. 2.

*Remark* 5.2. Brooks et al. (2019) only consider centering and biasing. Kobler et al. (2022b) use running mean for centering during the training. Kobler et al. (2022a) use different momentum to update running statistics for training and testing and multi-channel mechanisms for domain adaptation. Nevertheless, all of them are based on Eq. (19). Therefore, tricks such as multi-channel and separate momentum can also be applied to our GyroBN. This is what we mean by claiming that our GyroBN incorporates their approaches.

## 6 GYROBNS ON GRASSMANNIAN AND HYPERBOLIC SPACES

As indicated by Prop. 3.6 and Thm. 4.1, our GyroBN can be implemented on the Grassmannian and hyperbolic spaces, with the ability to normalize sample statistics. Our Alg. 1 allows us to implement GyroBN in a plug-and-play manner. This section clarifies additional technical details regarding its application in these two spaces. As Prop. 3.6 has demonstrated the equivalence of the gyrodistance and geodesic distance on these spaces, we use the terms "gyromean" and "gyrovariance" interchangeably with their Riemannian counterparts.

### 6.1 GRASSMANNIAN GYROBN

We focus on the ONB perspective. To the best of our knowledge, this is the first RBN on the Grassmannian. Given a batch of activations $\{U_{1\cdots N}\}$ over $\{\mathrm{Gr}(p,n), \oplus^{\mathrm{Gr}}\}$, Eq. (15) can be expressed as

$$\text{Centering to the identity } I_{p,n}: U_i^1 = \mathrm{mexp}\left(-[\overline{MM^\top}, \widetilde{I}_{p,n}]\right)U_i, \tag{20}$$

$$\text{Scaling the dispersion from } I_{p,n}: U_i^2 = \mathrm{mexp}\left(\frac{s}{\sqrt{v^2+\epsilon}}[\overline{U_i^1(U_i^1)^\top}, \widetilde{I}_{p,n}]\right)I_{p,n}, \tag{21}$$

$$\text{Biasing towards parameter } B \in \mathcal{M}: U_i^3 = \mathrm{mexp}\left([\overline{B}, \widetilde{I}_{p,n}]\right)U_i^2 \tag{22}$$

where $\overline{(\cdot)} = \widetilde{\mathrm{Log}}_{\widetilde{I}_{p,n}}(\cdot)$ with $\widetilde{\mathrm{Log}}$ as the Riemannian logarithm under the projector perspective, and $M$ is the Riemannian mean of $\{U_{1\cdots N}\}$, and $\widetilde{I}_{p,n} = I_{p,n}I_{p,n}^\top$ is the identity element under the projector perspective. The Riemannian mean can be calculated by the Karcher flow (Karcher, 1977). We use Alg. 5.3 by Bendokat et al. (2024) for a stable and efficient computation of the Riemannian logarithm required in the Karcher flow. For $[\overline{MM^\top}, \widetilde{I}_{p,n}]$ and $[\overline{U_i^1(U_i^1)^\top}, \widetilde{I}_{p,n}]$, inspired by Bendokat et al. (2024, Alg. 5.3) and Nguyen et al. (2024, Prop. 3.12), we have the following for fast computation.

**Proposition 6.1.** [↓] *Given* $U = (U_1^\top, U_2^\top)^\top \in \mathrm{Gr}(p,n)$ *with* $U_1 \in \mathbb{R}^{p\times p}$ *and* $U_2 \in \mathbb{R}^{(n-p)\times p}$, *then we have the following*

$$[\overline{UU^\top}, \widetilde{I}_{p,n}] = \begin{pmatrix} 0 & -\widetilde{U}_2^T \\ \widetilde{U}_2 & 0 \end{pmatrix}, \tag{23}$$

*where* $\widetilde{U}_2 = U_2 Q\frac{\arcsin(\hat{S})}{\hat{S}}R^\top$ *and* $U_1^\top \overset{\mathrm{SVD}}{:=} QSR^\top$. *Here* $S$ *is in ascending order,* $Q$ *and* $R$ *are column-wisely flipped accordingly, and* $\hat{S} = \sqrt{1-S^2}$.

*Remark* 6.2. Although we focus on the GyroBN under the ONB perspective, the GyroBN under the projector perspective can be calculated via the ONB perspective by the following process: (1) mapping data into the ONB perspective by $\pi^{-1} : \widetilde{\mathrm{Gr}}(p,n) \to \mathrm{Gr}(p,n)$; (2) normalizing data by the GyroBN under $\mathrm{Gr}(p,n)$; (3) mapping normalized data back to $\widetilde{\mathrm{Gr}}(p,n)$ by $\pi$. Technical details are presented in App. E.

### 6.2 HYPERBOLIC GYROBN

We focus on the Poincaré ball model over the hyperbolic space, *i.e.*, $\mathbb{P}_K^n$. The specific manifestation can be conducted in a plug-in manner. We simply need to plug the required operators from Tab. 2 and Tab. 8 into Alg. 1. Besides, the Poincaré Fréchet mean can be calculated by Lou et al. (2020, Alg. 1)].

*Remark* 6.3. As shown by Cannon et al. (1997), there are five isometric models that one can work for the hyperbolic spaces. Although we focus on the Poincaré ball, the GyroBN under other isometric metrics can also be easily constructed via the Poincaré GyroBN. The overall process is similar to Rmk. 6.2. For more detail, please refer to Lem. E.1 and Thm. E.2.

# 7 EXPERIMENTS

Our GyroBN layers are model-agnostic and can be seamlessly integrated into any network operating over the gyrospaces listed in Tab. 2. This section evaluates the effectiveness of our GyroBN on Grassmannian and hyperbolic neural networks.

## 7.1 EXPERIMENTS ON THE GRASSMANNIAN

**Implementation.** We focus on a newly developed Grassmannian network, GyroGr (Nguyen & Yang, 2023), which replaces the non-intrinsic transformation block (FRMap + ReOrth layers) in the classic GrNet (Huang et al., 2018) with Grassmannian left gyrotranslation. This modification has demonstrated improved numerical performance and stability (Nguyen & Yang, 2023). GyroGr is constructed over the ONB Grassmannian and consists of three basic blocks: left gyrotranslation, pooling (Huang et al., 2018), and the projection map (ProjMap) (Huang et al., 2018). The projection map functions as an activation layer that maps data into symmetric matrices for classification. Following Nguyen & Yang (2023), we evaluate our method on skeleton-based action recognition tasks, including the HDM05 (Müller et al., 2007), NTU60 (Shahroudy et al., 2016), and NTU120 (Liu et al., 2019) datasets, focusing on mutual actions for NTU60 and NTU120. For a fair comparison, we also extend ManifoldNorm (Chakraborty, 2020, Alg. 1-2) and RBN (Lou et al., 2020, Alg. 2) to the Grassmannian. Although these two BN methods were not originally implemented over the Grassmannian, they can be adapted by leveraging Riemannian operators such as geodesics, exponential and logarithmic maps, and parallel transport. *The key difference is that our GyroBN can normalize data distributions across different geometries, whereas the other two methods cannot.* Further details on datasets, implementation, and training efficiency are provided in App. F.2.

Table 3: Comparison of GyroBN against other Grassmannian BNs under GyroGr backbone.

| BN | None | | ManifoldNorm-Gr | | RBN-Gr | | GyroBN-Gr | |
|---|---|---|---|---|---|---|---|---|
| Acc. | Mean±std | Max | Mean±std | Max | Mean±std | Max | Mean±std | Max |
| HDM05 | 48.97±0.24 | 49.23 | 49.67±0.76 | 50.41 | 48.64±0.77 | 49.49 | **51.89±0.37** | **52.43** |
| NTU60 | 70.13±0.16 | 70.32 | 68.56±0.43 | 69.14 | 67.77±0.52 | 68.35 | **72.60±0.04** | **72.65** |
| NTU120 | 53.76±0.18 | 53.96 | 51.41±0.38 | 51.92 | 50.56±0.22 | 50.82 | **55.47±0.10** | **55.59** |

Table 4: Ablation of Grassmannian GyroBN under various network architectures.

| Architecture | HDM05 | | | | NTU60 | | | | NTU120 | | | |
|---|---|---|---|---|---|---|---|---|---|---|---|---|
| | 1Block | 2Block | 3Block | 4Block | 1Block | 2Block | 3Block | 4Block | 1Block | 2Block | 3Block | 4Block |
| GyroGr | 49.23 | 49.09 | 47.02 | 27.36 | 70.32 | 70.14 | 70.23 | 65.03 | 53.96 | 54.1 | 54.59 | 47.59 |
| GyroGrBN | **52.43** | **50.62** | **51.56** | **30.29** | **72.65** | **71.93** | **72.25** | **66.67** | **55.59** | **56.15** | **54.63** | **48.9** |

**Main results.** We compare our GyroBN with previous BN methods under the 1Block GyroGr backbone, which consists of one block of gyrotranslation and pooling layers followed by a ProjMap layer. We apply the BN after the pooling layer. The 5-fold results are presented in Tab. 3. GyroBN consistently delivers improved performance, enhancing the average performance of the vanilla GyroGr by 2.92%, 2.47%, and 1.71%. In contrast, Grassmannian ManifoldNorm and RBN could degrade the vanilla GyroGr network, particularly on the NTU60 and NTU120 datasets. This is primarily due to GyroBN's theoretical guarantee of normalizing sample statistics, a capability lacking in previous methods such as ManifoldNorm and RBN, as summarized in Tab. 1. Additionally, we observe that GyroBN mitigates the performance gap between training and testing, indicating its ability to enhance the model's generalization. Detailed discussions are provided in App. F.5. Overall, the above findings highlight the effectiveness of our GyroBN in facilitating network training.

**Ablations on the architecture.** We further validate our GyroBN across different architectures within the GyroGr baseline, which includes up to four blocks of gyrotranslation and pooling. GyroBN is applied after the first pooling layer, and we denote GyroGr with GyroBN as GyroGrBN. As implied by Tab. 3, both GyroGr and GyroGrBN exhibit relatively small variances, allowing us to conduct ablations using a single trial. Tab. 4 reports the results across all three datasets. We observe that GyroBN consistently improves the performance of the vanilla GyroGr baseline, highlighting the effectiveness of the GyroBN framework. Notably, as the network depth increases, the performance of the GyroGr backbone, with or without GyroBN, degrades. This is because the dimensionality of the

final feature becomes excessively low, leading to underfitting. For instance, in the deepest architecture with four blocks on the HDM05 dataset, the final dimension is $12 \times 10$, which is insufficient to effectively capture discriminative information. Despite this, our GyroBN still generally enhances the performance of the GyroGr backbone.

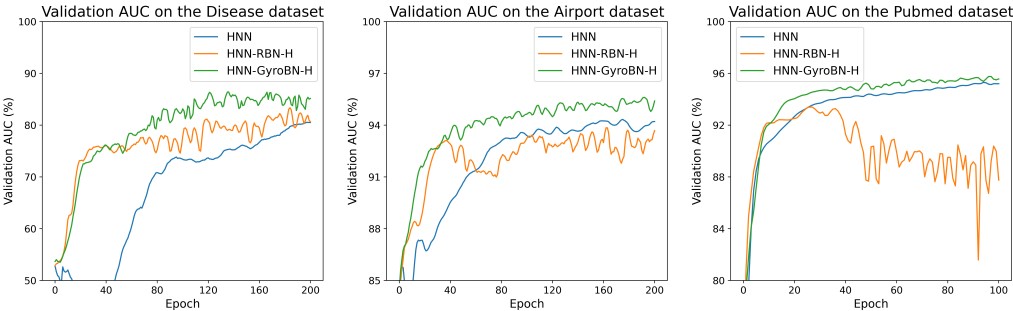

Figure 3: Validation AUC of HNN with or without GyroBNH or RBN-H on the Disease, Airport and Pubmed datasets. The results indicate that GyroBN facilitates the training of HNNs, improving both convergence and performance, and surpasses the previous RBN-H.

## 7.2 EXPERIMENTS ON THE HYPERBOLIC SPACE

**Implementation.** Following Lou et al. (2020), we validate our hyperbolic GyroBN, referred to as GyroBN-H, on the Hyperbolic Neural Network (HNN) (Ganea et al., 2018) backbone for the link prediction task using the Cora (Sen et al., 2008), Disease (Anderson & May, 1991), Airport (Zhang & Chen, 2018), and Pubmed (Namata et al., 2012) datasets. We also compare GyroBN-H with the hyperbolic RBN proposed by Lou et al. (2020, Alg. 2), denoted as RBN-H. Notably, the core difference between RBN and our GyroBN lies in the normalization – our GyroBN can normalize sample statistics across different geometries, while RBN lacks this guarantee. More details can be found in App. F.3.

Table 5: Comparison of HNN with or without GyroBN-H or RBN-H on the link prediction task.

| Dataset | HNN | HNN-RBN-H | HNN-GyroBN-H |
|---------|-----|-----------|--------------|
| Cora | 89.0 ± 0.1 | 93.5 ± 0.5 | **94.3 ± 0.2** |
| Disease | 75.1 ± 0.3 | 76.6 ± 2.2 | **81.2 ± 0.9** |
| Airport | 90.8 ± 0.2 | 94.2 ± 0.4 | **95.4 ± 0.2** |
| Pubmed | 94.9 ± 0.1 | 93.4 ± 0.2 | **95.8 ± 0.1** |

**Results.** Tab. 5 reports 5-fold average results of testing AUC on the four datasets. Although RBN and our GyroBN can generally improve the vanilla HNN, our GyroBN shows consistently better improvement than RBN. We also observe that RBN compromises HNN on the Pubmed dataset, while our GyroBN can consistently facilitate network training. Empowered by the normalization of sample statistics, our GyroBN also shows a smaller standard deviation than RBN. The above empirical results demonstrate the superiority of our GyroBN framework. Additionally, the validation AUC curves shown in Fig. 3 further highlight the effectiveness of our GyroBN in facilitating network training.

## 8 CONCLUSION

We propose a novel gyro structure on manifolds, termed the pseudo-reductive gyrogroup. This structure establishes the minimal prerequisites for developing valid normalization techniques over gyro spaces. Building on this gyro structure, we introduce a novel framework for batch normalization over gyrogroups, called GyroBN, which enables the normalization of non-Euclidean statistics. Several previous RBN methods, such as LieBN on general Lie groups (including three groups on SPD manifolds and the group of rotation matrices) and various AIM-based SPDBN methods, are revealed to be special cases of our GyroBN. Additionally, we manifest GyroBN on the Grassmannian and hyperbolic spaces. To the best of our knowledge, our Grassmannian GyroBN is the first RBN method on the Grassmannian manifold with a theoretical capability to normalize sample statistics. Experiments on Grassmannian and hyperbolic networks confirm the effectiveness of GyroBN. As a future avenue, we will implement our GyroBN over gyro spaces. We hope that our work will facilitate the development of deep networks for data with non-trivial geometries.

ACKNOWLEDGMENTS

This work was partly supported by the MUR PNRR project FAIR (PE00000013) funded by the NextGenerationEU, and the EU Horizon project ELIAS (No. 101120237). The authors also gratefully acknowledge the financial support from the China Scholarship Council (CSC), as well as the CINECA award under the ISCRA initiative for the availability of partial HPC resources support.

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

APPENDIX CONTENTS

## A  LIMITATION

As shown in Prop. 3.6, several geometries, including SPD, Grassmannian, hyperbolic, and hyperpherical manifolds, have gyro structures that can enable GyroBN with a theoretical control over sample statistics. However, some geometries do not have gyro structures, or their gyro structures are still untouched. Therefore, GyroBN cannot be established on these manifolds. We will explore other techniques to establish an RBN framework for these non-gyro geometries.

## B  NOTATIONS

Tab. 6 summarizes all the notations in the main paper.

Table 6: Summary of notations.

| Notation | Explanation |
|---|---|
| $\{G, \oplus\}$ or abbreviated as $G$ | A gyrogroup |
| $\{\mathcal{M}, \oplus, g\}$ or abbreviated as $\mathcal{M}$ | A Riemannian manifold $\{\mathcal{M}, g\}$ with a gyrogroup structure induced by $g$ |
| $\mathbb{1}$ | identity map |
| $T_P \mathcal{M}$ | Tangent space at $P \in \mathcal{M}$ |
| $g_p(\cdot, \cdot)$ or $\langle \cdot, \cdot \rangle_P$ | Riemannian metric at $P \in \mathcal{M}$ |
| $\|\cdot\|_P$ | The norm induced by $\langle \cdot, \cdot \rangle_P$ on $T_P \mathcal{M}$ |
| $\mathrm{d}_{\mathrm{geo}}(\cdot, \cdot)$ | Geodesic distance |
| $\mathrm{Log}_P$ | Riemannian logarithm at $P$ |
| $\mathrm{Exp}_P$ | Riemannian exponentiation at $P$ |
| $f_{*, P}$ | The differential map of the smooth map $f$ at $P \in \mathcal{M}$ |
| $\mathrm{PT}_{P \to Q}$ | Parallel transportation along the geodesic connecting $P$ and $Q$ |
| $E$ | Gyro identity of $\{\mathcal{M}, \oplus\}$ |
| $\ominus P$ | Group inverse of $P \in \mathcal{M}$ |
| $\odot$ | Gyro scalar product |
| $\mathrm{gyr}[\cdot, \cdot]$ | Gyration |
| $\langle \cdot, \cdot \rangle_{\mathrm{gr}}$ | Gyro inner product |
| $\|\cdot\|_{\mathrm{gr}}$ | Gyronorm |
| $\mathrm{d}_{\mathrm{gry}}(\cdot, \cdot)$ | Gyrodistance |
| FM | Fréchet mean under a gyrodistance |
| $\mathrm{Bar}_\eta(\cdot, \cdot)$ | Binary barycenter based on a gyrodistance |
| $\langle \cdot, \cdot \rangle$ | The standard Frobenius inner product |
| $\|\cdot\|$ | Norm induced by the standard Frobenius inner product |
| mlog | Matrix logarithm |
| mexp | Matrix exponentiation |
| $\mathscr{L}$ | Cholesky decomposition |
| Dlog | The diagonal element-wise logarithm |
| $\psi_{\mathrm{LC}}$ | Dlog $\circ \mathscr{L}$ |
| $\mathcal{S}^n_{++}$ | The SPD manifold |
| $\mathcal{S}^n$ | The Euclidean space of symmetric matrices |
| $\mathbb{P}^n_K$ | $n$-dimensional Poincaré ball with curvature $K < 0$ |
| $\mathbb{R}^n$ | $n$-dimensional Euclidean space |
| $\mathbb{D}^n_K$, | $n$-dimensional projected hypersphere with curvature $K > 0$ |
| $\mathcal{M}_K$ | Constant Curvature Spaces (CCS) |
| $\mathrm{Gr}(p, n)$ | The Grassmannian under the ONB perspective |
| $\widetilde{\mathrm{Gr}}(p, n)$ | The Grassmannian under the projector perspective |
| $I_{p,n}$ | The Grassmannian identity under the ONB perspective |
| $\widetilde{I}_{p,n}$ | The Grassmannian identity under the projector perspective |
| $\pi$ | The Riemannian isometry from $\mathrm{Gr}(p, n)$ onto $\widetilde{\mathrm{Gr}}(p, n)$ |
| $[\cdot, \cdot]$ | Matrix commutator |
| $[\cdot]$ | An element in $\mathrm{Gr}(p, n)$, which is a equivalent class |
| $I_n$ | $n \times n$ identity matrix |
| $P^\theta$ | Matrix power for SPD matrix $P$ |
| $\oplus^{\mathrm{AI}}, \oplus^{\mathrm{LE}}$ and $\oplus^{\mathrm{LC}}$ | Gyro additions on the SPD manifold under AIM, LEM and LCM |
| $\widetilde{\oplus}^{\mathrm{Gr}}$ and $\oplus^{\mathrm{Gr}}$ | Gyro additions on the Grassmannian under the ONB and projector perspectives |
| $\oplus_K$ | Gyro additions on the CCS |
| $\overline{(\cdot)}$ | $\overline{(\cdot)} = \widetilde{\mathrm{Log}}_{\widetilde{I}_{p,n}}(\cdot)$ with $\widetilde{\mathrm{Log}}$ as the Riemannian logarithm on $\widetilde{\mathrm{Gr}}(p, n)$ |
| $g^{\mathrm{AI}}, g^{\mathrm{LE}}$ and $g^{\mathrm{LC}}$ | AIM, LEM, LCM on the SPD manifold |
| $g^{\mathrm{Gr}}$ and $\widetilde{g}^{\mathrm{Gr}}$ | Riemannian metrics on the Grassmannian under the ONB and projector perspectives |
| $\tan_K$ | $\tan_K = \tan$ if $k > 0$, elif $K > 0$, $\tan_K = \tanh$ |

## C PRELIMINARIES

### C.1 RIEMANNIAN GEOMETRY

Manifolds can be intuitively understood as locally Euclidean spaces. Differentials generalize the concept of classical derivatives. For a detailed introduction to smooth manifolds, we refer readers to Tu (2011); Lee (2013). A Riemannian manifold is a manifold equipped with a Riemannian metric, which can be intuitively interpreted as a point-wise inner product. This metric allows for the adaptation of various Euclidean operators to the manifold setting. For an in-depth discussion on Riemannian manifolds, see Do Carmo & Flaherty Francis (1992); Lee (2018).

**Definition C.1** (Riemannian Manifolds (Do Carmo & Flaherty Francis, 1992)). A Riemannian metric on $\mathcal{M}$ is a smooth symmetric covariant 2-tensor field on $\mathcal{M}$, which is positive definite at every point. A Riemannian manifold is a pair $\{\mathcal{M}, g\}$, where $\mathcal{M}$ is a smooth manifold and $g$ is a Riemannian metric.

The isometries generalize the bijection in the set theory into the Riemannian geometry. If two manifolds are isometric, they can be viewed as equivalent. The Riemannian operators in these two manifolds are also closely related (Chen et al., 2024d, App. A. 2). The following defines the Riemannian isometry.

**Definition C.2** (Isometries (Lee, 2018)). If $\{\mathcal{M}, g\}$ and $\{\widetilde{\mathcal{M}}, \widetilde{g}\}$ are both Riemannian manifolds, a smooth map $f : \mathcal{M} \to \widetilde{\mathcal{M}}$ is called a (Riemannian) isometry if it is a diffeomorphism that satisfies

$$g_p(V, W) = \widetilde{g}_{f(p)}(f_{*,p}(V), f_{*,p}(W)), \tag{24}$$

where $f_{*,p}(\cdot) : T_p\mathcal{M} \to T_{f(p)}\widetilde{\mathcal{M}}$ is the differential map of $f$ at $p \in \mathcal{M}$, and $V, W \in T_p\mathcal{M}$ are two tangent vectors.

The exponential & logarithmic maps and parallel transportation are also crucial for Riemannian approaches in machine learning. To bypass the notation burdens caused by their definitions, we review the geometric reinterpretations of these operators (Pennec et al., 2006; Do Carmo & Flaherty Francis, 1992). In detail, in a manifold $\mathcal{M}$, geodesics correspond to straight lines in Euclidean space. A tangent vector $\overrightarrow{xy} \in T_x\mathcal{M}$ can be locally identified to a point $y$ on the manifold by geodesic starting at $x$ with an initial velocity of $\overrightarrow{xy}$, i.e. $y = \mathrm{Exp}_x(\overrightarrow{xy})$. On the other hand, the logarithmic map is the inverse of the exponential map, generating the initial velocity of the geodesic connecting $x$ and $y$, i.e., $\overrightarrow{xy} = \mathrm{Log}_x(y)$. These two operators generalize the idea of addition and subtraction in Euclidean space. For the parallel transportation $\mathrm{PT}_{x \to y}(V)$, it is a generalization of parallelly moving a vector along a curve in Euclidean space. we summarize the reinterpretation in Tab. 7.

Table 7: The geometric reinterpretations of Riemannian operators.

| Operations | Euclidean spaces | Riemannian manifolds |
|---|---|---|
| Straight line | Straight line | Geodesic |
| Subtraction | $\overrightarrow{xy} = y - x$ | $\overrightarrow{xy} = \mathrm{Log}_x(y)$ |
| Addition | $y = x + \overrightarrow{xy}$ | $y = \mathrm{Exp}_x(\overrightarrow{xy})$ |
| Parallelly moving | $V \to V$ | $\mathrm{PT}_{x \to y}(V)$ |

A Lie group is a manifold with a smooth group structure. It is a combination of algebra and geometry.

**Definition C.3** (Lie Groups). A manifold is a Lie group, if it forms a group with a group operation $\odot$ such that $m(x, y) \mapsto x \odot y$ and $i(x) \mapsto x_{\odot}^{-1}$ are both smooth, where $x_{\odot}^{-1}$ is the group inverse of $x$.

The following are some naive examples of the Lie group:

1. The set of real numbers $\mathbb{R}$, whose group operation is the addition.

2. The set of $n \times n$ invertible matrices $\mathrm{GL}(n)$, whose group operation is the matrix product. This group is known as the general linear group.

3. The set of $n \times n$ orthogonal matrices $\mathrm{O}(n)$, whose group operation is the matrix product. This group is a subgroup of $\mathrm{GL}(n)$, known as the orthogonal group.

## C.2 GYROVECTOR SPACES

Gyrogroups in Def. 2.1 generalize groups to non-associative algebraic systems by gyrations. Similarly, the gyrovector space generalizes the vector space, which has shown impressive success in hyperbolic geometry (Ungar, 2005; 2009; 2012; 2014).

**Definition C.4** (Gyrovector Spaces (Nguyen, 2022a))**.** A gyrocommutative gyrogroup $\{G, \oplus\}$ equipped with a scalar multiplication $\odot : \mathbb{R} \times G \to G$ is called a gyrovector space if it satisfies the following axioms for $s, t \in \mathbb{R}$ and $a, b, c \in G$:

(V1) $1 \odot a = a, 0 \odot a = t \odot e = e,$ and $(-1) \odot a = \ominus a$.

(V2) $(s + t) \odot a = s \odot a \oplus t \odot a$.

(V3) $(st) \odot a = s \odot (t \odot a)$.

(V4) $\mathrm{gyr}[a, b](t \odot c) = t \odot \mathrm{gyr}[a, b]c$.

(V5) $\mathrm{gyr}[s \odot a, t \odot a] = \mathbb{1}$, where $\mathbb{1}$ is the identity map.

Gyrovector spaces generalize vector spaces to curved geometries, such as the SPD and Grassmannian manifolds (Nguyen, 2022a). While retaining familiar properties like distributivity (V2) and associativity (V3), they incorporate the complexities of gyrations.

## C.3 MATRIX AND VECTOR MANIFOLDS

### C.3.1 DEFINITIONS

**SPD:** The set $\mathcal{S}_{++}^n$ of $n \times n$ SPD matrices form a manifold, named the SPD manifold (Pennec et al., 2006). We focus on three popular Riemannian metrics on the SPD manifold: Affine-Invariant Metric (AIM) (Pennec et al., 2006), Log-Euclidean Metric (LEM) (Arsigny et al., 2005), and Log-Cholesky Metric (LCM) (Lin, 2019).

**Grassmannian:** The Grassmannian is the set of $p$-dimensional subspace of $n$-dimensional vector space (Tu, 2011), which has two matrix representations (Bendokat et al., 2024):

$$\begin{aligned}
\text{Projector perspective: } \widetilde{\mathrm{Gr}}(p, n) &= \{P \in \mathcal{S}^n : P^2 = P, \mathrm{rank}(P) = p\}, \\
\text{ONB perspective: } \mathrm{Gr}(p, n) &= \{[U] : [U] := \{\widetilde{U} \in \mathrm{St}(p, n) \mid \widetilde{U} = UR, R \in \mathrm{O}(p)\}\},
\end{aligned} \tag{25}$$

where $\mathcal{S}^n$ is the Euclidean space of symmetric matrices, $\mathrm{St}(p, n)$ is the Stiefel manifold, and $\mathrm{O}(p)$ is the orthogonal group. By abuse of notations, we use $[U]$ and $U$ interchangeably for the element of $\mathrm{Gr}(p, n)$. As shown by Helmke & Moore (2012), the ONB perspective $\mathrm{Gr}(p, n)$ is diffeomorphism to $\widetilde{\mathrm{Gr}}(p, n)$ by

$$\pi(U) = UU^\top, \forall U \in \mathrm{Gr}(p, n), \tag{26}$$

where the $n \times p$ column-wise orthonormal matrix $U$ should be more precisely understood as a representative of an equivalence class (Bendokat et al., 2024).

**CCS:** The Poincaré ball $\mathbb{P}_K^n$ for the hyperbolic space (Ungar, 2009; Ganea et al., 2018), projected hypersphere $\mathbb{D}_K^n$ for the hypersphere (Skopek et al., 2019), and standard Euclidean space $\mathbb{R}^n$ (Zorich & Paniagua, 2016) are more generally called the Constant Curvature Space (CCS) (Do Carmo & Flaherty Francis, 1992), as they have constant sectional curvature $K$. The $n$-dimensional Poincaré ball and projected hypersphere are represented as $\mathbb{P}_K^n = \{\boldsymbol{x} \in \mathbb{R}^n : \langle x, x \rangle < -\frac{1}{K}\}$ with $K < 0$ and $\mathbb{D}_K^n = \mathbb{R}^n$ with $K > 0$. When $K = 0$, the CCS becomes the standard Euclidean space $\mathbb{R}^n$.

### C.3.2 GYRO AND RIEMANNIAN STRUCTURES

For a matrix manifold $\mathcal{M}$ and a CCS $\mathcal{M}_K$, we make the following notations:

$$\{\mathcal{M}, g\} = \begin{cases} \{\mathcal{S}_{++}^n, g^{\mathrm{AI}}\} & \text{(The SPD manifold under AIM)} \\ \{\mathcal{S}_{++}^n, g^{\mathrm{LE}}\} & \text{(The SPD manifold under LEM)} \\ \{\mathcal{S}_{++}^n, g^{\mathrm{LC}}\} & \text{(The SPD manifold under LEM)} \\ \{\mathrm{Gr}(p, n), g^{\mathrm{Gr}}\} & \text{(The Grassmannian under the ONB perspective)} \\ \{\widetilde{\mathrm{Gr}}(p, n), \widetilde{g}^{\mathrm{Gr}}\} & \text{(The Grassmannian under the projector perspective)} \end{cases} \tag{27}$$

$$\mathcal{M}_K = \begin{cases} \mathbb{P}_K^n, & \text{for } K < 0 \\ \mathbb{R}^n, & \text{for } K = 0 \\ \mathbb{D}_K^n, & \text{for } K > 0 \end{cases} \tag{28}$$

$$\tan_K = \begin{cases} \tan & \text{if } K > 0 \\ \tanh & \text{if } K < 0 \end{cases} \tag{29}$$

**Notations in Tab. 2:** We summarize all the necessary group operations in Tab. 2 with the following notations. Given any $P, Q \in \mathcal{M}$ with $\mathcal{M}$ as $\mathcal{S}_{++}^n$ or $\widetilde{\mathrm{Gr}}(p, n)$, and $x, y \in \mathcal{M}_K$ with $\mathcal{M}_K$ as $\mathbb{P}_K^n (K < 0)$, $\mathbb{D}_K^n (K > 0)$ or $\mathbb{R}^n (K = 0)$, we make the following notations. For the Grassmannian, $U = \pi^{-1}(P)$ and $V = \pi^{-1}(Q)$ are the ONB representations. We denote matrix exponential, matrix logarithm, and Cholesky decomposition as $\mathrm{mexp}$, $\mathrm{mlog}$, and $\mathscr{L}$, respectively. We denote $\psi_{\mathrm{LC}} = \mathrm{Dlog} \circ \mathscr{L}$, where $\psi_{\mathrm{LC}}$ is the diagonal logarithm. As shown by Chen et al. (2024e), LCM and the associated Lie group on $\mathcal{S}_{++}^n$ are pulled back by $\psi_{\mathrm{LC}}$ for the Euclidean space of lower triangular matrices. $I_n$ is the $n \times n$ identity matrix, $I_{p,n} = (I_p, 0)^\top \in \mathbb{R}^{n \times p}$, and $\widetilde{I}_{p,n} = \pi(I_{p,n})$. For the Grassmannian $\widetilde{\mathrm{Gr}}(n, p)$, $\Omega = [\overline{P}, \widetilde{I}_{p,n}]$, where $\overline{P} = \mathrm{Log}_{\widetilde{I}_{p,n}}(P)$ and $[\cdot, \cdot]$ is the matrix commutator. We denote $\langle \cdot, \cdot \rangle$ and $\| \cdot \|$ as the standard (matrix & vector) inner product and norm.

We further make the following notations for the related Riemannian operators in Tab. 8. Given $P \in \mathcal{M}$ ($x \in \mathcal{M}_K$), we denote the tangent vector as $V \in T_P\mathcal{M}$ ($v \in T_x\mathcal{M}_K$). We denote the geodesic distance, Riemannian logarithm, and Riemannian exponential as $\mathrm{d}_{\mathrm{geo}}$, $\mathrm{Log}$ and $\mathrm{Exp}$, respectively.

Table 8: Riemannian geometries of several matrix and vector manifolds. For CCS $\mathcal{M}_K$, we present the operators for $K \neq 0$, as when $K = 0$, all the Riemannian operators are reduced to the familiar vector operators.

| Manifolds | $\mathrm{d}_{\mathrm{geo}}(P, Q)$ or $\mathrm{d}_{\mathrm{geo}}(x, y)$ | $\mathrm{Log}_P Q$ or $\mathrm{Log}_x y$ | $\mathrm{Exp}_P V$ or $\mathrm{Exp}_x v$ | References |
|---|---|---|---|---|
| $\{\mathcal{S}_{++}^n, g^{\mathrm{LE}}\}$ | $\|\mathrm{mlog}(P) - \mathrm{mlog}(Q)\|$ | $(\mathrm{mlog}_{*,P})^{-1}(\mathrm{mlog}(Q) - \mathrm{mlog}(P))$ | $\mathrm{mexp}(\mathrm{mlog}(P) + \mathrm{mlog}_{*,P}(V))$ | (Arsigny et al., 2005) |
| $\{\mathcal{S}_{++}^n, g^{\mathrm{AI}}\}$ | $\left\|\mathrm{mlog}\left(Q^{-\frac{1}{2}}PQ^{-\frac{1}{2}}\right)\right\|$ | $P^{\frac{1}{2}}\mathrm{mlog}\left(P^{-\frac{1}{2}}QP^{-\frac{1}{2}}\right)P^{\frac{1}{2}}$ | $P^{\frac{1}{2}}\mathrm{mexp}\left(P^{-\frac{1}{2}}VP^{-\frac{1}{2}}\right)P^{\frac{1}{2}}$ | (Pennec et al., 2006) |
| $\{\mathcal{S}_{++}^n, g^{\mathrm{LC}}\}$ | $\|\psi_{\mathrm{LC}}(P) - \psi_{\mathrm{LC}}(Q)\|$ | $(\mathscr{L}^{-1})_{*,L}(\psi_{\mathrm{LC}}(Q) - \psi_{\mathrm{LC}}(P))$ | $\psi_{\mathrm{LC}}^{-1}(\psi_{\mathrm{LC}}(P) + (\psi_{\mathrm{LC}})_{*,P}(V))$ | (Lin, 2019) (Chen et al., 2024e) |
| $\mathrm{Gr}(p, n)$ | $\|\mathrm{arccos}(\Sigma)\|$ $P^\top Q \overset{\mathrm{SVD}}{:=} O\Sigma R^\top$ | $O\arctan(\Sigma)R^\top$ $(I_n - PP^\top)Q(P^\top Q)^{-1} \overset{\mathrm{SVD}}{:=} O\Sigma R^\top$ | $[PR\cos(\Sigma)R^T + O\sin(\Sigma)R^T]$ $V \overset{\mathrm{SVD}}{:=} O\Sigma R^\top$ | (Edelman et al., 1998) (Bendokat et al., 2024) |
| $\widetilde{\mathrm{Gr}}(p, n)$ | $\frac{1}{\sqrt{2}}\|\mathrm{mlog}((I_n - 2Q)(I_n - 2P))\|$ | $\frac{1}{2}[\mathrm{mlog}((I_n - 2Q)(I_n - 2P)), P]$ | $\mathrm{mexp}([V, P])P\,\mathrm{mexp}(-[V, P])$ | (Bendokat et al., 2024) (Batzies et al., 2015) |
| $\mathcal{M}_K$ | $\frac{2}{\sqrt{|K|}}\tan_K^{-1}\left(\sqrt{|K|}\|-x \oplus_K y\|\right)$ | $\frac{2}{\sqrt{|K|\lambda_x^K}}\tan_K^{-1}\left(\sqrt{|K|}\|-x \oplus_K y\|\right)\frac{-x \oplus_K y}{\|-x \oplus_K y\|}$ | $x \oplus_K\left(\tan_K\left(\sqrt{|K|}\frac{\lambda_x^K\|v\|}{2}\right)\frac{v}{\sqrt{|K|}\|v\|}\right)$ | (Do Carmo & Flaherty Francis, 1992) (Petersen, 2006) (Skopek et al., 2019) |

*Remark* C.5 (The Grassmannian and cut locus). Due to the cut locus of the Grassmannian, the logarithm map does not exist globally (Bendokat et al., 2024). In this paper, when we use $\mathrm{Log}_P(Q)$ on the Grassmannian, we implicitly assume $P$ and $Q$ are not in each other's cut locus. Besides, more precisely, the gyro addition and scalar multiplication on Grassmannian are also not globally defined (Nguyen, 2022a), due to the cut locus. Following Nguyen (2022a) and Nguyen & Yang (2023), we implicitly assume the gyro operations are well-defined on the Grassmannian.

### C.4 INTUITIVE EXPLANATIONS OF GYROGROUPS

This subsection intuitively explains gyrogroups by contrast with the trivial $\mathbb{R}$. Gyrogroups are proposed to generalize the concept of addition in $\mathbb{R}$ or $\mathbb{R}^n$ to non-Euclidean spaces.

**Gyrogroups in Def. 2.1** extend the concept of groups to non-associative algebraic systems, *i.e.*, $a \oplus (b \oplus c) \neq (a \oplus b) \oplus c$. The gyrogroups over the manifolds in Tab. 2 are defined by Eqs. (3), (5) and (6).

More importantly, these definitions are natural generalizations of Euclidean vector operations. Take the gyro addition Eq. (3) as an example. When the manifold $\mathcal{M}$ is the Euclidean space $\mathbb{R}^n$, the gyro addition defined by Eq. (3) becomes the familiar vector addition.

**Gyration & gyroassociativity:** The gyration is an automorphism $\mathrm{gyr}[a, b] : G \to G$ for each $a, b \in G$. It is called an automorphism as it can preserve the gyro addition:

$$\mathrm{gyr}[a, b](c \oplus d) = \mathrm{gyr}[a, b](c) \oplus \mathrm{gyr}[a, b](d), \quad \forall c, d \in G. \tag{30}$$

The gyration is used to generalize the associativity:

$$a + (b + c) = (a + b) + c, \quad \forall a, b, c \in \mathbb{R}. \tag{31}$$

Take the Grassmannian gyrogroup $\{\mathrm{Gr}(p, n), \oplus^{\mathrm{Gr}}\}$ as an example. For any $U, V, R \in \oplus^{\mathrm{Gr}}$, we have

$$\text{Non-associativity: } V \oplus^{\mathrm{Gr}} (U \oplus^{\mathrm{Gr}} R) \neq (V \oplus^{\mathrm{Gr}} U) \oplus^{\mathrm{Gr}} R, \tag{32}$$

$$\text{Left gyroassociativity: } V \oplus^{\mathrm{Gr}} (U \oplus^{\mathrm{Gr}} R) = (V \oplus^{\mathrm{Gr}} U) \oplus^{\mathrm{Gr}} \mathrm{gyr}^{\mathrm{Gr}}[U, V](R). \tag{33}$$

Eq. (33) differs with Eq. (31) only in a gyration. The concrete expression of the gyration over the Grassmannian is presented by Nguyen (2022a, Def. 3.18).

**Left reduction** is defined as $\mathrm{gyr}[a, b] = \mathrm{gyr}[a \oplus b, b]$ for any $a$ and $b$ in the gyrogroup $G$. It is called a "left reduction" because $b$ in $a \oplus b$ can be canceled out.

**Gyrocommutativity in Def. 2.2** generalizes the commutative property $(a + b = b + a)$ by the gyration operator $\mathrm{gyr}[a, b]$.

**Nonreductive gyrogroups in Def. 2.3** are relaxed gyrogroups, allowing structures where the left reduction law (G4) does not hold. Its prototype comes from the gyrogroup of the Grassmannian (Nguyen, 2022a, Thm. 3.20), where (G4) does not hold.

**Relation between left reduction, non-reduction, & pseudo-reduction:** The left reduction can induce many basic properties of gyrogroups ( see Thm. D.1 and Ungar (2009, Thms. 1.13 and 1.14)). However, non-reduction does not guarantee several basic properties. This drawback will undermine the rationality of the non-reductive gyrogroups. Therefore, as an intermediate, we propose pseudo-reduction. The associated pseudo-reductive gyrogroups maintain most of the basic properties of gyrogroups. Please refer to Thm. D.1 and its remark for more details.

## D FIRST PSEUDO-REDUCTIVE GYROGROUPS PROPERTIES

**Theorem D.1** (First Pseudo-reductive Gyrogroups Properties)**.** *Let $\{G, \oplus\}$ be a pseudo-reductive gyrogroup. For any elements $P, Q, R, X \in G$, we have:*

1. *If $P \oplus Q = P \oplus R$, then $Q = R$ (General Left Cancellation law; see (9) below).*

2. $\mathrm{gyr}[E, P] = \mathbb{1}$ *for any left identity $E$ in $G$.*

3. $\mathrm{gyr}[X, P] = \mathbb{1}$ *for any left inverse $X$ of $P$ in $G$.*

4. *There is $P$ left identity which is $P$ right identity.*

5. *There is only one left identity.*

6. *Every left inverse is $P$ right inverse.*

7. *There is only one left inverse, $\ominus P$, of $P$, and $\ominus(\ominus P) = P$.*

8. *The left cancellation law: $\ominus P \oplus (P \oplus Q) = Q$.*

9. *The gyrator identity: $\mathrm{gyr}[P, Q]X = \ominus(P \oplus Q) \oplus \{P \oplus (Q \oplus X)\}$.*

10. $\mathrm{gyr}[P, Q]E = E$.

11. $\mathrm{gyr}[P, Q](\ominus X) = \ominus \mathrm{gyr}[P, Q]X$.

12. $\mathrm{gyr}[P, E] = \mathbb{1}$.

13. *The gyrosum inversion law: $\ominus(P \oplus Q) = \mathrm{gyr}[P, Q](\ominus Q \ominus P)$*

**Credit of the proof.** Most of the proof is borrowed from Ungar (2009, Thms. 1.13 and 1.14), which presents the corresponding properties for gyrogroups. After re-analyzing these two theorems, we find that most of the proof can be readily extended into pseudo-reductive gyrogroups.

*Proof.* This theorem follows Ungar (2009, Thms. 1.13 and 1.14), which presents some useful properties for gyrogroups. We argue that all the properties except $\text{gyr}[a, a] = \mathbb{1}$ are independent of the left reduction law (G4), and are therefore satisfied on pseudo-reductive gyrogroups. All the properties can be proven in the same way as the ones for Thms. 1.13 and 1.14 by Ungar (2009). We summarize the logic in the following:

- left gyroassociativity $\Rightarrow$ 1
- left gyroassociativity + 1 $\Rightarrow$ 2
- definition $\Rightarrow$ 3
- left gyroassociativity + 1 + 3 $\Rightarrow$ 4
- definition $\Rightarrow$ 5
- left gyroassociativity + (G2) + 1 + 3 + 4 + 5 $\Rightarrow$ 6
- 1+6 $\Rightarrow$ 7
- left gyroassociativity +3 $\Rightarrow$ 8
- left gyroassociativity + left cancellation in 8 $\Rightarrow$ 9
- gyro identity in 9 $\Rightarrow$ 10
- 10 $\Rightarrow$ 11
- left cancellation in 8 + gyro identity in 9 $\Rightarrow$ 12
- left cancellation in 8 + gyro identity in 9 $\Rightarrow$ 13

$\square$

*Remark* D.2. For non-reductive gyrogroups, 2 and 3 are agnostic. Therefore, every property based on 2 or 3 is not guaranteed, such as 4, and from 6 to 10. The missing of these basic properties will undermine the rationality of non-reductive gyrogroups. In contrast, most basic properties of gyrogroups are preserved in our pseudo-reductive gyrogroups.

## E  GRASSMANNIAN GYROBN UNDER THE PROJECTOR PERSPECTIVE

Given two isometric manifolds $\{\mathcal{M}_1, g^1\}$ and $\{\mathcal{M}_2, g^2\}$, the induced gyro structures in Eq. (3)-Eq. (9) have the following relations. We first present a useful lemma.

**Lemma E.1.** *Given manifolds $\{\mathcal{M}_1, g^1\}$ and $\{\mathcal{M}_2, g^2\}$ and a Riemannian isometry $f : \mathcal{M}_1 \to \mathcal{M}_2$, we have the following:*

1. *The groupoid $\{\mathcal{M}_1, \oplus_1\}$ induced by $g^1$ is pseudo-reductive (left-invariant), iff the groupoid $\{\mathcal{M}_2, \oplus_2\}$ induced by $g^2$ is pseudo-reductive (left-invariant);*

2. *Any gyration in $\{\mathcal{M}_1, g^1\}$ preserves gyronorm iff any gyration in $\{\mathcal{M}_2, g^2\}$ preserves gyronorm;*

3. *$f$ preserves the gyrodistance.*

*Proof.* First, we review some facts about gyrogroups under the Riemannian isometry. As shown by Nguyen & Yang (2023, Thm. 2.5), $\{\mathcal{M}_1, \oplus_1\}$ satisfies (G1-G3) in Def. 2.1 iff $\{\mathcal{M}_2, \oplus_2\}$ satisfies (G1-G3). Besides, $f^{-1}$ is an isomorphism satisfying

$$f^{-1}(P \oplus_2 Q) = f^{-1}(P) \oplus_1 f^{-1}(Q), \tag{34}$$

$$f^{-1}(t \odot_2 P) = t \oplus_1 f^{-1}(P), \tag{35}$$

$$f^{-1}(\ominus_2 P) = \ominus_1 f^{-1}(P), \tag{36}$$

$$\text{gyr}_2[P, Q](R) = f\left(\text{gyr}_1[f^{-1}(P), f^{-1}(Q)](f^{-1}(R))\right). \tag{37}$$

where $P, Q, R \in \mathcal{M}_2$ are arbitrary points, $t \in \mathbb{R}$ is a real scalar, $\ominus_i$, $\mathrm{gyr}_i$ are the gyro inverses and gyrations on $\mathcal{M}_i$ for $i = 1, 2$. $f$ is also an isomorphism with similar properties.

We only need to prove one direction for the iff condition for the pseudo-reduction or norm invariance. We focus on $\Rightarrow$ and follow the above notations in the following.

**Pseudo-reduction:**

$$
\begin{aligned}
\mathrm{gyr}_2[\ominus P, P] &= f \circ \mathrm{gyr}_1[f^{-1}(\ominus_2 P), f^{-1}(P)] \circ f^{-1} \text{ (Eq. (37))} \\
&= f \circ \mathrm{gyr}_1[\ominus_1 f^{-1}(P), f^{-1}(P)] \circ f^{-1} \text{ (Eq. (36))} \\
&= \mathbb{1}.
\end{aligned}
\tag{38}
$$

**Gyronorm invariance under gyrations:** For simplicity, we denote the gyronorm, identity element, and Riemannian logarithm on $\mathcal{M}_i$ as $\|\|_i$ $E_i$, and $\mathrm{Log}^i$, respectively. First, the following demonstrates that the Riemannian isometry $f$ preserves gyronorm:

$$
\begin{aligned}
\|P\|_2 &= \left\|\mathrm{Log}^2_{E_2}(P)\right\|_{E_2} \\
&\overset{(1)}{=} \left\|\mathrm{Log}^1_{f^{-1}(E_2)}\left(f^{-1}(P)\right)\right\|_{f^{-1}(E_2)} \\
&\overset{(2)}{=} \left\|\mathrm{Log}_{E_1}\left(f^{-1}(P)\right)\right\|_{E_1} \\
&= \left\|f^{-1}(P)\right\|_1
\end{aligned}
\tag{39}
$$

(1) As $f : \mathcal{M}_1 \to \mathcal{M}_2$ is a Riemannian isometry, we have the following equations:

$$
\mathrm{Log}^2_P Q = f_{*, f^{-1}(P)}\left(\mathrm{Log}^1_{f^{-1}(P)}\left(f^{-1}(Q)\right)\right), \forall P, Q \in \mathcal{M}_2,
\tag{40}
$$

$$
g^2_P(V, W) = g^1_{f^{-1}(P)}((f^{-1})_{*, P}(V), (f^{-1})_{*, P}(W)), \forall V, W \in T_P \mathcal{M}_2,
\tag{41}
$$

where $(\cdot)_*$ is the differential map;

(2) $E_2 = f(E_1)$.

Then we have the following:

$$
\begin{aligned}
\|\mathrm{gyr}_2[P, Q](R)\|_2 &\overset{(1)}{=} \left\|f\left(\mathrm{gyr}_1[f^{-1}(P), f^{-1}(Q)](f^{-1}(R))\right)\right\|_2 \\
&\overset{(2)}{=} \left\|\mathrm{gyr}_1[f^{-1}(P), f^{-1}(Q)](f^{-1}(R))\right\|_1 \\
&\overset{(3)}{=} \left\|f^{-1}(R)\right\|_1 \\
&\overset{(4)}{=} \|R\|_2
\end{aligned}
\tag{42}
$$

The above derivation comes from the following.

(1) Eq. (37);

(2) Eq. (39);

(3) Any gyration on $\mathcal{M}_1$ can preserve the gyronorm;

(4) Eq. (39).

**Invariance of gyrodistance under $f$:** Denoting the gyrodistance on $\mathcal{M}_i$ as $\mathrm{d}_i$, for any $U, V \in \mathcal{M}_1$, we have the following:

$$
\begin{aligned}
\mathrm{d}_1(U, V) &= \|\ominus_1 U \oplus_1 V\|_1 \\
&\overset{(1)}{=} \|f(\ominus_1 P \oplus_1 Q)\|_2 \\
&\overset{(2)}{=} \|\ominus_1 f(P) \oplus_2 f(Q)\|_2 \\
&= \mathrm{d}_2(f(P), f(Q))_2
\end{aligned}
\tag{43}
$$

The above derivation comes from the following.

(1) $f$ preserves gyronorm;

(2) $f$ is an isomorphism.

□

Given a batch of activations $\{P_{1\ldots N}\}$ on a gyrogroup $\{\mathcal{M}, \oplus\}$, we denote the GyroBN as

$$\text{GyroBN}(\{P_i\}; B, s, \epsilon, \eta), \tag{44}$$

where $B \in \mathcal{M}$ and $s$ are biasing and scaling parameters, $\epsilon$ is a small positive value, and $\eta$ is the momentum.

**Theorem E.2.** *Given manifolds $\{\mathcal{M}_1, g^1\}$ and $\{\mathcal{M}_2, g^2\}$ and a Riemannian isometry $f : \mathcal{M}_1 \to \mathcal{M}_2$, for a batch of activation $\{P_{1\ldots N}\}$ in $\mathcal{M}_1$, $\text{GyroBN}_1(P_i; B, s, \epsilon, \gamma)$ in $\mathcal{M}_1$ can be calculated as*

$$\text{GyroBN}_1(P_i; B, s, \epsilon, \gamma) = f^{-1}\left(\text{GyroBN}_2(f(P_i); f(B), s, \epsilon, \gamma)\right), \tag{45}$$

*where $\text{GyroBN}_2$ is the GyroBN in $\mathcal{M}_2$.*

*Proof.* This theorem is inspired by Chen et al. (2024b, thm. 5.3), which characterizes the LieBNs under isometric manifolds. As gyrogroups are natural generalizations of Lie groups, our GyroBN is expected to have similar results. The following proof follows a similar logic to the one by Chen et al. (2024b, thm. 5.3), except that all operations are gyro operations.

For $i = 1, 2$, we denote Eq. (15) and binary gyro barycenter on $\mathcal{M}_i$ as $\xi^i(\cdot|M, v^2, B, s)$ and $\text{Bar}_\eta^i(\cdot, \cdot)$. Let $\mathcal{B} = \{P_{1\ldots N}\}$ and $f(\mathcal{B}) = \{f(P_{1\ldots N})\}$. We only need to show the following:

$$M_2 = f(M_1), v_1 = v_2, \tag{46}$$

$$\xi^1(P_i|M_1, v^2, B, s) = f^{-1}(\xi^2(f(P_i)|M_2, v^2, f(B), s)), \tag{47}$$

$$\text{Bar}_\eta^1(P, Q) = f^{-1}\left(\text{Bar}_\eta^2(f(P), f(Q))\right), \forall P, Q \in \mathcal{M}_1. \tag{48}$$

where $M_i$ and $v_i$ are the batch Fréchet mean and variance over $\mathcal{M}_i$ for $i = 1, 2$. Eqs. (46) and (48) can be directly obtained by the invariance of gyrodistance under $f$ (Lem. E.1). We only need to show Eq. (47).

We have the following:

$$\begin{aligned}
f^{-1}(\xi^2(f(P_i)|M_2, v^2, f(B), s)) &= f^{-1}(f(B) \oplus_2 (t \odot_2 (\ominus_2 M_2 \oplus_2 f(P_i)))) \\
&\overset{(1)}{=} f^{-1} \circ f\left(B \oplus_1 (t \odot (\ominus_1 M_1 \oplus_1 P_i))\right) \\
&= B \oplus_1 (t \odot (\ominus_1 M_1 \oplus_1 P_i)) \\
&= \xi^1(P_i|M, v^2, B, s),
\end{aligned} \tag{49}$$

where $t = \frac{s}{\sqrt{v^2 + \epsilon}}$. The above derivation comes from the following.

(1) $f$ is an isomorphism preserving gyro operations.

□

As $\pi^{-1} : \widetilde{\text{Gr}}(p, n) \to \text{Gr}(p, n)$ is a Riemannian isometry, Thm. E.2 indicates that the GyroBN under the projector perspective can be calculated by the ONB perspective by the following process:

1. mapping data into the ONB perspective by $\pi^{-1} : \widetilde{\text{Gr}}(p, n) \to \text{Gr}(p, n)$;

2. normalizing data by the GyroBN under $\text{Gr}(p, n)$;

3. mapping normalized data back to $\widetilde{\text{Gr}}(p, n)$ by $\pi$.

Besides, both Lem. E.1 or Prop. 3.6 can guarantee theoretical control over the gyromean and gyrovariance under the projector perspective.

## F  EXPERIMENTAL DETAILS AND ADDITIONAL ABLATIONS

### F.1  SUMMARY OF OPERATORS IN GRASSMANNIAN AND HYPERBOLIC GYROBNS

The specific implementation of Alg. 1 on a gyrogroup can be carried out in a plug-in manner. This involves simply substituting the required operators from Tabs. 2 and 8 into Alg. 1. To streamline this process, we summarize the discussion in Sec. 6 in Tab. 9, where we present all the required operators for the Grassmannian and hyperbolic GyroBNs.

Table 9: Key operators in calculating GyroBN on the Grassmannian and hyperbolic manifolds. Here $P, Q \in \mathrm{Gr}(p, n)$ are two ONB Grassmannian points, while $x, y \in \mathbb{P}^n_K$ are two Poincaré vectors. Other notations follow from Tabs. 2 and 8.

| Operator | $\mathrm{Gr}(p, n)$ | $\mathbb{P}^n_K$ |
|---|---|---|
| Identity element | $I_{p,n}$ | $0 \in \mathbb{R}^n$ |
| $P \oplus^{\mathrm{Gr}} Q$ or $x \oplus_K y$ | $\mathrm{mexp}(\Omega)V$ | $\frac{\left(1-2K\langle x,y\rangle - K\|y\|^2\right)x + \left(1+K\|x\|^2\right)y}{1-2K\langle x,y\rangle + K^2\|x\|^2\|y\|^2}$ |
| $\ominus^{\mathrm{Gr}} P$ or $\ominus_K x$ | $\mathrm{mexp}(-\Omega)I_{p,n}$ | $-x$ |
| $t \odot^{\mathrm{Gr}} P$ or $t \odot_K x$ | $\mathrm{mexp}(t\Omega)I_{p,n}$ | $\frac{1}{\sqrt{|K|}} \tanh\left(t\tanh^{-1}(\sqrt{|K|}\|x\|)\right)\frac{x}{\|x\|}$ |
| $\mathrm{Bar}^{\mathrm{Gr}}_\gamma(Q, P)$ or $\mathrm{Bar}^K_\gamma(y, x)$ | $\mathrm{Exp}^{\mathrm{Gr}}_P(\gamma \mathrm{Log}^{\mathrm{Gr}}_P(Q))$ | $x \oplus_K (-x \oplus_K y) \odot_K t$ |
| Fréchet Mean | Karcher Flow (Karcher, 1977) | (Lou et al., 2020, Alg. 1) |

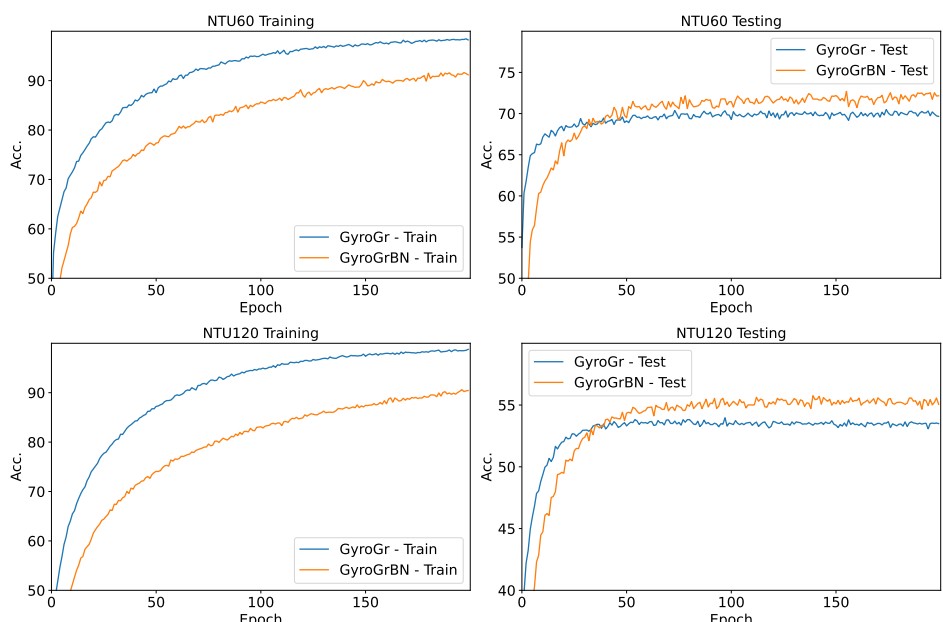

Figure 4: Training and testing performance on the NTU datasets of 1Block GyroGr. Our BN improves the generalization abilities of GyroGr.

## F.2 DETAILS ON THE GRASSMANNIAN EXPERIMENTS

### F.2.1 DATASETS AND PREPROCESSING

**HDM05[3] (Müller et al., 2007).** It consists of 2,273 skeleton-based motion capture sequences executed by different actors. Each frame consists of 3D coordinates of 31 joints. We remove the under-represented clips, trimming the dataset down to 2086 instances scattered throughout 117 classes. Following Nguyen & Yang (2023), we model each sequence as a $93 \times 10$ Grassmannian matrix.

**NTU60 (Shahroudy et al., 2016).** It has 56,880 sequences of 3D skeleton data classified into 60 classes, where each frame contains the 3D coordinates of 25 or 50 body joints. We use mutual actions and follow the cross-view protocol (Shahroudy et al., 2016). Following Nguyen & Yang (2023), we model each sequence as a $150 \times 10$ Grassmannian matrix.

**NTU120[4] (Liu et al., 2019).** This dataset contains 114,480 sequences in 120 action classes. We use mutual actions and follow the cross-setup protocol (Liu et al., 2019). Following Nguyen & Yang (2023), we model each sequence as a $150 \times 10$ Grassmannian matrix.

---

[3]https://resources.mpi-inf.mpg.de/HDM05/
[4]https://github.com/shahroudy/NTURGB-D

### F.2.2 IMPLEMENTATION DETAILS

**Architectures.** The 1Block GyroGr is structured as: gyrotranslation → pooling → [GyroBN] → ProjMap → classification. The KBlock GyroGr consists of $K$ blocks of gyrotranslation and pooling, followed by a final ProjMap layer for classification. Since each pooling operation halves the dimensionality, we omit pooling in the last block when $K > 1$. Following Huang et al. (2018), the number of channels is set to 8. The momentum for GyroBN is the default value of 0.1. Additionally, in line with Nguyen & Yang (2023), we use the Cayley transform to approximate matrix exponentiation over skew-symmetric matrices.

**Optimization.** Following Nguyen & Yang (2023), we adopt the trivialization tricks (Lezcano Casado, 2019) for the Grassmannian parameters in the gyrotranslation and our GyroBN layers. Given a parameter $U \in \mathrm{Gr}(p, n)$, we parameterize it by a matrix $\mathfrak{U} \in \mathbb{R}^{(n-p) \times p}$ such that

$$\begin{bmatrix} 0 & -\mathfrak{U}^T \\ \mathfrak{U} & 0 \end{bmatrix} = \left[ \overline{UU^\top}, \widetilde{I}_{p,n} \right]. \tag{50}$$

The biasing parameter $U$ can be retrieved by

$$U = \mathrm{mexp}\left( \left[ \overline{UU^\top}, \widetilde{I}_{p,n} \right] \right) I_{p,n} = \mathrm{mexp}\left( \begin{bmatrix} 0 & -\mathfrak{U}^T \\ \mathfrak{U} & 0 \end{bmatrix} \right) I_{p,n}. \tag{51}$$

In this way, all parameters lie in Euclidean spaces. We, therefore, can directly use the optimizers in Pytorch (Paszke et al., 2019). In all experiments, we use an SGD optimizer with a learning rate of 5e-2 and zero weight decay. The batch size is 30, and training epochs are 400, 200, and 200 for the HDM05, NTU60, and NTU120 datasets.

### F.3 DETAILS ON THE HYPERBOLIC EXPERIMENTS

#### F.3.1 DATASETS

**Cora (Sen et al., 2008).** It is a citation network where nodes represent scientific papers in the area of machine learning, edges are citations between them, and node labels are academic (sub)areas.

**Disease (Anderson & May, 1991).** It represents a disease propagation tree, simulating the SIR disease transmission model, with each node representing either an infection or a non-infection state.

**Airport (Zhang & Chen, 2018).** It is a transductive dataset where nodes represent airports and edges represent the airline routes as from OpenFlights.org.

**Pubmed (Namata et al., 2012).** This is a standard benchmark describing citation networks where nodes represent scientific papers in the area of medicine, edges are citations between them, and node labels are academic (sub)areas.

#### F.3.2 IMPLEMENTATION DETAILS

We use the official implementations of HGCN[5] (Chami et al., 2019) and Lou et al. (2020)[6] to conduct the experiments. We follow all the settings as Lou et al. (2020, Sec. H.1) to implement our experiments. Specifically, the baseline encoder is an HNN (Ganea et al., 2018) with two layers: the first maps the input feature dimension to 128, and the second maps 128 to 128. Hyperbolic BN layers, such as GyroBN or RBN, follow each layer. We use the Adam optimizer (Kingma, 2014), with a learning rate of $1e^{-2}$ and a weight decay of $1e^{-3}$, except for the Cora dataset, where the weight decay is set to 0.

### F.4 HARDWARE

All experiments except the ones on the Pubmed dataset use an Intel Core i9-7960X CPU with 32GB RAM and an NVIDIA GeForce RTX 2080 Ti GPU. The experiments on the Pubmed dataset are conducted on a single NVIDIA Quadro RTX A6000 48GB GPU.

### F.5 GYROBN AND MODEL GENERALIZATION ABILITY

We denoted GyroGr with our GyroBN as GyroGrBN. The training and testing curves on the NTU datasets are presented in Fig. 4. We can observe that the gap between training and testing performance is much lower when GyroGr is endowed with the Grassmannian GyroBN. This phenomenon indicates that our GyroBN can improve the generalization ability of the Grassmannian neural networks.

---

[5] https://github.com/HazyResearch/hgcn
[6] https://github.com/CUAI/Differentiable-Frechet-Mean

## F.6 Training efficiency

### F.6.1 Training efficiency on the Grassmannian networks

Table 10: Average training time (seconds/epoch) for GyroGr with and without different Grassmannian BNs. The dimensions of the input Grassmannian matrices for the BN layer are also reported.

| Methods | HDM05 $(47 \times 10)$ | NTU60 $(75 \times 10)$ | NTU120 $(75 \times 10)$ |
|---|---|---|---|
| GyroGr | 2.19 | 50.92 | 80.72 |
| GyroGr-ManifoldNorm | 4.98 | 242.12 | 409.48 |
| GyroGr-RBN | 5.16 | 242.63 | 410.08 |
| GyroGr-GyroBN | 3.10 | 59.55 | 108.92 |

As the Riemannian computations over the ONB Grassmannian $\mathrm{Gr}(p, n)$ involve matrix decompositions, the disparity in efficiency is more evident.

**Empirical Results.** Tab. 10 shows the training efficiency of GyroGr with and without different Grassmannian BNs. Compared to other Grassmannian BNs, GyroBN is significantly more efficient due to the simplicity of the gyro operation.

**Analysis.** The key difference between GyroBN (Alg. 1) and the ManifoldNorm (Chakraborty, 2020, Algs. 1-2) or RBN (Lou et al., 2020, Alg. 2) lies in their methods for centering, biasing, and scaling. GyroBN uses gyro operations, while ManifoldNorm and RBN rely on Riemannian operators, such as parallel transport and the Riemannian logarithmic and exponential maps. This distinction underpins the efficiency of GyroBN. The three primary contributing factors, ranked by importance, are as follows:

1. **Riemann vs. Gyro.** The Riemannian operators over the Grassmannian involve computationally expensive processes like SVD decomposition or matrix inversion (see Tab. 8 for Riemannian exp and log, and (Edelman et al., 1998, Thm. 2.4) for parallel transport). Consequently, ManifoldNorm and RBN require multiple SVD or matrix inversion operations. In contrast, GyroBN is relatively simpler. As discussed in Sec. 6.1, the gyro operation can be further simplified, and the involved SVD is performed on a reduced $p \times p$ matrix instead of $n \times p$. Additionally, as noted in App. F.2.2, computationally intensive matrix exponentiation is efficiently approximated using the Cayley map.

2. **Reduced Matrix Products.** As shown in Tab. 8, each Riemannian operator involves several matrix products over $n \times p$ matrices. GyroBN reduces these to matrix products over $(n - p) \times p$ or $p \times p$ matrices, which are computationally more efficient, such as Prop. 6.1.

3. **Optimization.** The biasing parameter optimization in GyroBN is simpler. ManifoldNorm uses an $n \times n$ orthogonal matrix for biasing, and RBN employs an $n \times p$ Grassmannian matrix, both requiring Riemannian optimization. In contrast, GyroBN applies trivialization tricks (parameterizing manifold data via the Riemannian exp (Lezcano Casado, 2019), as shown in Eq. (51)), making the biasing parameter a $(n-p) \times p$ Euclidean matrix. Furthermore, Eq. (51) and Eq. (22) share a similar form, allowing them to be jointly simplified. Although RBN could adopt similar trivialization for biasing, this would introduce an additional Riemannian exp step, leaving little advantage over the Riemannian optimization. In summary, GyroBN benefits from joint simplification with trivialization, whereas the other two Grassmannian BN methods require additional Riemannian optimization.

### F.6.2 Training efficiency on the hyperbolic networks

Table 11: Average training time (seconds/epoch) for HNN with and without different hyperbolic BNs.

| Methods | Cora | Disease | Airport | Pubmed |
|---|---|---|---|---|
| HNN | 0.0323 | 0.0271 | 0.054 | 0.1253 |
| HNN-RBN-H | 0.0905 | 0.0883 | 0.1215 | 0.3416 |
| HNN-GyroBN-H | 0.0757 | 0.0842 | 0.119 | 0.3351 |

Recalling Tab. 8, the Riemannian logarithmic and exponential maps over the hyperbolic space contain gyro operations. Therefore, it is expected that our GyroBN is more efficient than RBN. Tab. 11 reports the average training time per epoch, indicating that our GyroBN is more efficient than RBN (Lou et al., 2020) in the hyperbolic space.

### F.7 EXPERIMENTS ON OTHER HYPERBOLIC BASELINES

The subsection validates our GryoBN on the HNN++ (Shimizu et al., 2020) and RResNet (Katsman et al., 2024) backbones. We follow the HNN experiments and use these two backbones as the encoder for the link prediction task.

### F.7.1 EXPERIMENTS ON HNN++

Table 12: Comparison of HNN++ with or without RBN-H and GyroBN-H on the Cora, Disease, and Airport datasets.

| Dataset | Cora | Disease | Airport |
|---|---|---|---|
| HNN++ | 91.06 ± 0.47 | 77.83 ± 1.39 | 94.93 ± 0.23 |
| HNN++-RBN-H | 88.22 ± 0.67 | 59.50 ± 1.67 | 94.91 ± 0.23 |
| HNN++-GyroBN-H | **92.83 ± 0.32** | **79.53 ± 0.74** | **95.89 ± 0.10** |

We evaluate HNN++ with and without RBN-H (Lou et al., 2020) and our GyroBN-H on the Cora, Disease, and Airport datasets. Like the HNN experiments, we used a two-layer HNN++ backbone as the encoder, with hyperbolic hidden feature dimensions set to 128. Tab. A reports the five-fold average results, leading to the following findings:

1. **GyroBN can improve and stabilize HNN++.** While RBN can degrade HNN++'s performance, particularly on the Cora dataset, GyroBN consistently enhances it. Additionally, we observed that HNN++ could be sensitive to weight decay. For instance, on the Cora dataset, setting the weight decay to $1e^{-3}$ reduces the performance to 62.55, prompting us to use zero weight decay for HNN++ on this dataset. In contrast, HNN++ with GyroBN is robust to weight decay. Furthermore, HNN++-GyroBN achieves the smallest performance variance across all three datasets, indicating that GyroBN facilitates more robust learning.

2. **GyroBN can accelerate convergence.** On the Cora dataset, HNN++ requires over 400 epochs to converge. In contrast, HNN++-GyroBN converges in approximately 150 epochs, significantly improving convergence.

### F.7.2 EXPERIMENTS ON RRESNET

Table 13: Comparison of RResNet with or without RBN-H and GyroBN-H under different hidden feature dimensions on the Cora dataset.

| Dim | 8 | 16 | 32 | 64 | 128 |
|---|---|---|---|---|---|
| RResNet | 71.77 ± 6.67 | 77.38 ± 10.82 | 77.14 ± 7.48 | 66.73±11.85 | 80.75±4.12 |
| RResNet-RBN-H | 61.94 ± 2.28 | 63.54 ± 3.01 | 62.26 ± 3.48 | 60.38±2.97 | 87.92±2.67 |
| RResNet-GyroBN-H | **84.58 ± 5.44** | **87.67 ± 2.53** | **88.08 ± 2.90** | **86.50±1.59** | **89.52±3.42** |

We focus on the RResNet Horo (Katsman et al., 2024), which leverages a vector field induced by the horosphere projection feature map. Following the settings of HNN and HNN++, we use 2 residual blocks as the backbone. For a more comprehensive comparison, we evaluate hidden feature dimensions ranging from 8 to 128. Tab. 13 reports the five-fold average results, leading to the following findings:

1. **GyroBN consistently improves performance.** While RBN performs well only under the 128-dimensional setting, it fails under other dimensions, performing worse than the baseline RResNet. In contrast, GyroBN consistently improves RResNet's performance across all hidden feature dimensions.

2. **GyroBN stabilizes performance and preserves representation power.** Both RResNet and RResNet-RBN exhibit significant fluctuations in performance across different hidden

dimensions. However, RResNet-GyroBN shows a more stable performance. This indicates that our GyroBN could maintain the representation power of the backbone network.

## F.8 EXPERIMENTS ON COVARIATE SHIFTS

The covariate shift serves as the main motivation for the classical Euclidean BN (Ioffe & Szegedy, 2015). Similar challenges arise in Riemannian networks, where the complexity of the manifold often disrupts data distributions, underscoring the utility of our proposed GyroBN. To demonstrate covariate shifts, we perform numerical experiments on the Grassmannian and hyperbolic networks.

### F.8.1 COVARIATE SHIFTS ON THE GRASSMANNIAN NETWORK

Table 14: Ten-fold results for geodesic distance $d(M_{out}, I_{p,n})$ and shift $\Delta = \frac{d(M_{out}, I_{p,n})}{b} \times 100$, where $b = \sqrt{p}\frac{\pi}{2}$ is the boundary of geodesic distance over $\mathrm{Gr}(p, n)$.

|  | Pooling | Transformation |
|---|---|---|
| $d(M_{out}, I_{p,n})$ | $2.90 \pm 0.11$ | $4.18 \pm 0.04$ |
| $\Delta$ (%) | $58.47\% \pm 2.28\%$ | $84.22\% \pm 0.72\%$ |

The transformation and pooling layers in the GyroGr baseline can be expressed as

$$f_{\mathrm{trans}} : \mathrm{Gr}(p, n) \to \mathrm{Gr}(p, n), \tag{52}$$
$$f_{\mathrm{pooling}} : \mathrm{Gr}(p, n) \to \mathrm{Gr}(p, n/2). \tag{53}$$

For simplicity, we assume $n$ is even for the pooling layer. Since the ONB Grassmannian $\mathrm{Gr}(p, n)$ is a quotient manifold and pooling changes dimensions, we use the geodesic distance between the batch mean and the identity element $I_{p,n} = (I_p, \mathbf{0})^\top \in \mathbb{R}^{n \times p}$ as a consistent measure. Note that the geodesic distance on $\mathrm{Gr}(p, n)$ is bounded by $\sqrt{p}\frac{\pi}{2}$ (Wong, 1967, Thm. 8).

We randomly generate 30 Grassmannian matrices of size $100 \times 10$ with an initial batch mean as the identity element $I_{p,n}$. We denote the resulting batch mean after transformation or pooling as $M_{out}$. If the distribution were preserved, the geodesic distance $d(M_{out}, I_{p,n})$ (or $d(M_{out}, I_{p,n/2})$ for pooling) would be zero. Tab. 14 shows that $M_{out}$ significantly deviates from $I_{p,n}$, indicating the covariate shift.

### F.8.2 COVARIATE SHIFTS ON THE HYPERBOLIC NETWORK

We examine the covariate shift in the transformation layer (Ganea et al., 2018, Lem. 6) in HNN. We focus on the canonical Poincaré ball (curvature $K = -1$). We randomly generate 30 5-dimensional Poincaré vectors. The batch mean vectors of input and output in the transformation layer are

$$M_{in} = [-0.0055, 0.0503, -0.0913, -0.0493, 0.0652]^\top, \tag{54}$$
$$M_{out} = [-0.0969, -0.0443, 0.0385, 0.0936, -0.0770]^\top. \tag{55}$$

The geodesic distance between $M_{in}$ and $M_{out}$ is 0.55, indicating the covariate shift in HNN.

## F.9 EXPERIMENTS ON CONDITION NUMBERS

In the GyroGr network, the transformation layer outputs $n \times p$ Grassmannian matrices, whose condition numbers are trivial since all singular values of a Grassmannian matrix are 1. Similarly, for the HNN model, the transformation layer outputs a vector, leading to a trivial condition number as well. Thus, we focus on evaluating the condition numbers of the weight matrices in the hidden transformation layer and the overall network Jacobian matrix. We evaluate Grassmannian GyroGr networks without loss of generality under one block architecture, both with and without different BN layers, using each trained model. The results demonstrate that GyroBN consistently reduces the condition numbers of the weight matrices and the network Jacobian.

### F.9.1 CONDITION NUMBERS OF WEIGHT MATRICES OF THE TRANSFORMATION LAYER

As shown in Tab. 15, GyroBN consistently reduces the condition numbers of the weight matrices across datasets. Interestingly, although ManifoldNorm achieves the lowest condition numbers, its performance could be even worse than the GyroGr baseline, suggesting that excessively reducing the condition number could limit the model's capacity to capture sufficient expressive features.

Table 15: Statistics of condition numbers for the weight matrices in the GyroGr transformation layer along with different normalization methods. As weight matrices have 8 channels, we report the mean, standard deviation (std), minimum, and maximum values of their condition numbers. The dimensions of the 8-channel transformation matrices are $83 \times 10$ on the HDM05 dataset and $140 \times 10$ on the other datasets.

| Dataset | BN | Mean | Min | Max |
|---------|-----|------|-----|-----|
| HDM05 | None | $3.81 \pm 0.23$ | 3.46 | 4.17 |
| | ManifoldNorm-Gr | $1.97 \pm 0.15$ | 1.83 | 2.30 |
| | RBN-Gr | $3.49 \pm 0.46$ | 2.96 | 4.43 |
| | GyroBN-Gr | $2.37 \pm 0.18$ | 2.12 | 2.67 |
| NTU60 | None | $2.77 \pm 0.12$ | 2.61 | 2.94 |
| | ManifoldNorm-Gr | $1.76 \pm 0.10$ | 1.63 | 1.92 |
| | RBN-Gr | $2.17 \pm 0.20$ | 1.94 | 2.51 |
| | GyroBN-Gr | $2.02 \pm 0.14$ | 1.82 | 2.28 |
| NTU120 | None | $3.35 \pm 0.28$ | 2.97 | 3.72 |
| | ManifoldNorm-Gr | $1.91 \pm 0.10$ | 1.80 | 2.14 |
| | RBN-Gr | $2.22 \pm 0.11$ | 2.00 | 2.36 |
| | GyroBN-Gr | $2.16 \pm 0.11$ | 2.00 | 2.33 |

### F.9.2 CONDITION NUMBER OF NETWORK JACOBIAN

We further analyze the condition number of the network Jacobian on the HDM05 and NTU120 datasets. We randomly select 100 samples to calculate the statistics of the associated network condition numbers, as shown in Tab. 16. Our GyroBN consistently lowers condition numbers compared to other methods across both datasets, indicating that GyroBN stabilizes network training. Additionally, the reduced maximum condition numbers, especially on the NTU120 dataset, suggest that GyroBN avoids extreme outliers in conditioning, further highlighting its advantage over previous methods. Notably, on the HDM05 dataset, both ManifoldNorm and RBN increase network conditioning.

Table 16: Condition numbers of network Jacobian (output w.r.t. input) across different normalization methods on the GyroGr baseline. The statistics come from 100 random samples. The dimensions of Jacobian matrices are $117 \times 930$ on the HDM05 dataset and $11 \times 9600$ on the NTU120 dataset.

| Dataset | BN | Mean | Min | Max |
|---------|-----|------|-----|-----|
| HDM05 | None | $61.05 \pm 4.15$ | 52.19 | 74.29 |
| | ManifoldNorm-Gr | $67.38 \pm 6.82$ | 51.15 | 88.81 |
| | RBN-Gr | $64.16 \pm 6.44$ | 50.54 | 80.07 |
| | GyroBN-Gr | $59.85 \pm 4.33$ | 48.46 | 70.47 |
| NTU120 | None | $122.24 \pm 68.45$ | 61.8 | 399.28 |
| | ManifoldNorm-Gr | $116.75 \pm 65.17$ | 75.46 | 443.01 |
| | RBN-Gr | $108.75 \pm 52.27$ | 63.19 | 349.25 |
| | GyroBN-Gr | $97.01 \pm 41.88$ | 61.19 | 262.55 |

## G PROOFS

### G.1 PROOF OF PROP. 3.2

*Proof of Prop. 3.2.* By Nguyen & Yang (2023, Lem. 2.3), easy computations show that Eq. (12) holds for $\mathrm{Gr}(p, n)$ iff it holds for $\widetilde{\mathrm{Gr}}(p, n)$. Without loss of generality, we prove the case for the projector perspective.

Given any $P, Q \in \widetilde{\mathrm{Gr}}(p, n)$, Def. 3.18 by Nguyen (2022a) gives the expression for gyration:

$$\mathrm{gyr}[\ominus P, P]Q = F(\ominus P, P)Q\left(F(\ominus P, P)\right)^{-1}, \tag{56}$$

with $F(\ominus P, P)$ defined as

$$F(\ominus P, P) = \mathrm{mexp}\left(-\left[\overline{\ominus P \oplus P}, \widetilde{I}_{p,n}\right]\right)\mathrm{mexp}\left(\left[\overline{\ominus P}, \widetilde{I}_{p,n}\right]\right)\mathrm{mexp}\left(\left[\overline{P}, \widetilde{I}_{p,n}\right]\right), \quad (57)$$

where $\overline{(\cdot)} = \mathrm{Log}_{\widetilde{I}_{p,n}}(\cdot)$. This equation can be further simplified as

$$\begin{aligned}
F(\ominus P, P) &\overset{(1)}{=} \mathrm{mexp}(0)\,\mathrm{mexp}\left(\left[\overline{\ominus P}, \widetilde{I}_{p,n}\right]\right)\mathrm{mexp}\left(\left[\overline{P}, \widetilde{I}_{p,n}\right]\right), \\
&\overset{(2)}{=} \mathrm{mexp}\left(\left[\overline{\ominus P}, \widetilde{I}_{p,n}\right]\right)\mathrm{mexp}\left(\left[\overline{P}, \widetilde{I}_{p,n}\right]\right), \\
&\overset{(3)}{=} I_n.
\end{aligned} \quad (58)$$

The above derivation follows from

(1) $\overline{\ominus P \oplus P} = \overline{\widetilde{I}_{p,n}} = 0 \in \mathbb{R}^{n \times n}$.

(2) $\mathrm{mexp}(0) = I_n$.

(3) $\overline{\ominus P} = -\overline{P}$ and $\mathrm{mexp}\left(\left[-\overline{P}, \widetilde{I}_{p,n}\right]\right) = \mathrm{mexp}\left(\left[\overline{P}, \widetilde{I}_{p,n}\right]\right)^{-1}$.

Therefore, $\mathrm{gyr}[\ominus P, P]$ is the identity map. $\qquad \square$

### G.2 PROOF OF THM. 3.3

*Proof of Thm. 3.3.* We first prove a useful lemma.

**Lemma G.1** (Left Gyrotranslation Law). *Every pseudo-reductive gyrogroups $\{G, \oplus\}$ verifies the left gyrotranslation law:*

$$\ominus(P \oplus Q) \oplus (P \oplus R) = \mathrm{gyr}[P, Q]\,(\ominus Q \oplus R), \forall P, Q, R \in G. \quad (59)$$

*Proof.* This lemma generalizes Lems. I.1 and L.1 by Nguyen & Yang (2023), which proves the left gyrotranslation law on the specific gyrogroups of the SPD and Grassmannian manifolds. Their proof only relies on the left cancellation and the basic axioms (G1-3). Note that the original proof of left gyrotranslation on the Grassmannian (Nguyen & Yang, 2023, Lems. I.1) is questionable, as relies on the left cancellation of gyrogroups, and the Grassmannian is not a gyrogroup but a non-reductive gyrogroup. Fortunately, as we show in Thm. D.1, the general pseudo-reductive gyrogroups, including the Grassmannian, enjoy left cancellation. Therefore, all the proof by Nguyen & Yang (2023, Lems. I.1) can be readily generalized into the general pseudo-reductive gyrogroups. $\qquad \square$

$\Rightarrow$: For any $R, S \in G$, the gyroautomorphism can be expressed by the gyrator identity in Thm. D.1:

$$\mathrm{gyr}[P, Q]R = X \oplus \bar{R}, \quad (60)$$

$$\mathrm{gyr}[P, Q]S = X \oplus \bar{S}, \quad (61)$$

where $X = \ominus(P \oplus Q)$, $\bar{R} = P \oplus (Q \oplus R)$, and $\bar{S} = P \oplus (Q \oplus S)$. Then Eq. (30) in (Nguyen & Yang, 2023) for the specific Grassmannian can be directly extended into the pseudo-reductive gyrogroup, as it only relies on left gyrotranslation, invariance of the norm under gyroautomorphisms, and the axioms of (G1-G3).

$\Leftarrow$:

$$\begin{aligned}
\|\mathrm{gyr}[P, Q](R)\|_{\mathrm{gr}} &= \|\mathrm{gyr}[P, Q](\ominus E \oplus R)\|_{\mathrm{gr}}, \quad (7 \text{ in Thm. D.1 indicates } \ominus E = E) \\
&= \|\ominus \mathrm{gyr}[P, Q](E) \oplus \mathrm{gyr}[P, Q](R)\|_{\mathrm{gr}}, \quad (\text{automorphism}) \\
&= \mathrm{d}_{\mathrm{gry}}\left(\mathrm{gyr}[P, Q](E), \mathrm{gyr}[P, Q](R)\right), \\
&= \mathrm{d}_{\mathrm{gry}}\left(E, R\right), \\
&= \|\ominus E \oplus R\|_{\mathrm{gr}}, \\
&= \|R\|_{\mathrm{gr}}.
\end{aligned} \quad (62)$$

$\qquad \square$

### G.3 PROOF OF THM. 3.5

*Proof of Thm. 3.5.* Given any $P, Q, R \in G$, we make the following proof.

**Gyroisometry of the left gyrotranslation:** This property generalizes Thms. 2.12 and 2.16 by Nguyen & Yang (2023), which deal with the gyrotranslations in the SPD and Grassmannian manifolds. We have the following:

$$
\begin{aligned}
\mathrm{d}_{\mathrm{gry}}(L_P(Q), L_P(R)) &= \mathrm{d}_{\mathrm{gry}}(P \oplus Q, P \oplus R), \\
&= \|\ominus(P \oplus Q) \oplus (P \oplus R)\|_{\mathrm{gr}}, \\
&= \|\mathrm{gyr}[P, Q](\ominus Q \oplus R)\|_{\mathrm{gr}} \text{ (left gyrotranslation)}, \\
&= \|\ominus Q \oplus R\|_{\mathrm{gr}} \text{ (gyroisometry of the automorphism)}, \\
&= \mathrm{d}_{\mathrm{gry}}(Q, R).
\end{aligned}
\tag{63}
$$

**Symmetry of the gyrodistance:**

$$
\begin{aligned}
\mathrm{d}_{\mathrm{gry}}(P, Q) &= \|\ominus P \oplus Q\|_{\mathrm{gr}}, \\
&= \|\ominus(\ominus P \oplus Q)\|_{\mathrm{gr}} \text{ (Eq. (5) and Eq. (7))}, \\
&= \|\mathrm{gyr}[\ominus P, Q](\ominus Q \oplus P)\|_{\mathrm{gr}} \text{ (gyrosum inversion law)}, \\
&= \|\ominus Q \oplus P\|_{\mathrm{gr}} \text{ (gyroisometry of the automorphism)}, \\
&= \mathrm{d}_{\mathrm{gry}}(Q, P).
\end{aligned}
\tag{64}
$$

**Gyroisometry of the gyroinverse:**

$$
\begin{aligned}
&\mathrm{d}_{\mathrm{gry}}(\ominus P, \ominus Q) \\
&= \|P \ominus Q\|_{\mathrm{gr}}, \\
&= \|\ominus Q \oplus P\|_{\mathrm{gr}} \text{ ( gyrocommutativity and gyroisometry of the automorphism)}, \\
&= \mathrm{d}_{\mathrm{gry}}(Q, P), \\
&= \mathrm{d}_{\mathrm{gry}}(P, Q) \text{ (Symmetry of the gyrodistance)}.
\end{aligned}
\tag{65}
$$

$\square$

### G.4 PROOF OF PROP. 3.6

*Proof of Prop. 3.6.* We first show the expressions of the gyrodistance on $\mathcal{M}$. Then we proceed to show the gyrodistance and gyroisometries on $\mathcal{M}_K$. We follow all the notations in Tab. 2 and denote the geodesic distance under a specific geometry as $\mathrm{d}_{\mathrm{geo}}$.

**Expressions of gyrodistance on $\mathcal{M}$:**

For $\{\mathcal{S}_{++}^n, \oplus^{\mathrm{AI}}\}$, we have the following:

$$
\begin{aligned}
\mathrm{d}_{\mathrm{gry}}(P, Q) &= \left\| P^{-\frac{1}{2}} Q P^{-\frac{1}{2}} \right\|_{\mathrm{gr}}, \\
&= \left\| \mathrm{mlog}(P^{-\frac{1}{2}} Q P^{-\frac{1}{2}}) \right\| (\mathrm{Log}_I = \mathrm{mlog}), \\
&= \mathrm{d}_{\mathrm{geo}}(P, Q).
\end{aligned}
\tag{66}
$$

For $\{\mathcal{S}_{++}^n, \oplus^{\mathrm{LE}}\}$, we have the following:

$$
\begin{aligned}
\mathrm{d}_{\mathrm{gry}}(P, Q) &= \|\mathrm{mexp}(-\mathrm{mlog}(P) + \mathrm{mlog}(Q))\|_{\mathrm{gr}}, \\
&= \left\| \mathrm{mlog}_{*,I}^{-1}(-\mathrm{mlog}(P) + \mathrm{mlog}(Q)) \right\|_I, (\mathrm{Log}_I(P) = \mathrm{mlog}_{*,I}^{-1}(\mathrm{mlog}(P))), \\
&= \|\mathrm{mlog}(P) - \mathrm{mlog}(Q)\|, \text{ (the pullback of LEM (Chen et al., 2024e))} \\
&= \mathrm{d}_{\mathrm{geo}}(P, Q),
\end{aligned}
\tag{67}
$$

where $\mathrm{mlog}_{*,I}^{-1}$ is the inverse of the differential map of $\mathrm{mlog}$, and $\|\cdot\|_I$ is the norm induced by the LEM at $I$.

As LCM is also a pullback metric (Chen et al., 2024e), $\{\mathcal{S}_{++}^n, \oplus^{\mathrm{LC}}\}$ follow the same logic:

$$
\begin{aligned}
\mathrm{d}_{\mathrm{gry}}(P, Q) &= \left\| \psi_{\mathrm{LC}}^{-1}(-\psi_{\mathrm{LC}}(P) + \psi_{\mathrm{LC}}(Q)) \right\|_{\mathrm{gr}}, \\
&= \left\| (\psi_{\mathrm{LC}})_{*,I}^{-1}(-\psi_{\mathrm{LC}}(P) + \psi_{\mathrm{LC}}(Q)) \right\|_I \quad (\mathrm{Log}_I(P) = (\psi_{\mathrm{LC}})_{*,I}^{-1}\psi_{\mathrm{LC}}(P)), \\
&= \left\| \psi_{\mathrm{LC}}(P) - \psi_{\mathrm{LC}}(Q) \right\|, \quad \text{(the pullback of LCM (Chen et al., 2024e))} \\
&= \mathrm{d}_{\mathrm{geo}}(P, Q).
\end{aligned}
\tag{68}
$$

For $\{\widetilde{\mathrm{Gr}}(p, n), \widetilde{\oplus}^{\mathrm{Gr}}\}$, we have the following:

$$
\begin{aligned}
\mathrm{d}_{\mathrm{gry}}(P, Q) &= \left\| \mathrm{mexp}(-[\overline{P}, \widetilde{I}_{p,n}]) Q \, \mathrm{mexp}([\overline{P}, \widetilde{I}_{p,n}]) \right\|_{\mathrm{gr}} \quad (\ominus P = -\overline{P}), \\
&= \left\| \widetilde{P}^\top Q \widetilde{P} \right\|_{\mathrm{gr}} \quad (\widetilde{P} = \mathrm{mexp}([\overline{P}, \widetilde{I}_{p,n}]) \in \mathrm{SO}(n)), \\
&= \frac{1}{2} \left\| \mathrm{Log}_{\widetilde{I}_{p,n}} \left( \widetilde{P}^\top Q \widetilde{P} \right) \right\| \quad (\|\cdot\|_I = \frac{1}{2} \|\cdot\|), \\
&= \frac{1}{2} \left\| [\Omega, \widetilde{I}_{p,n}] \right\|,
\end{aligned}
\tag{69}
$$

where $[\Omega, \widetilde{I}_{p,n}] = \mathrm{Log}_{\widetilde{I}_{p,n}} \left( \widetilde{P}^\top Q \widetilde{P} \right)$. For $\Omega$, we have the following:

$$
\begin{aligned}
\Omega &= \frac{1}{2} \mathrm{mlog} \left( \left( I_n - 2\widetilde{P}^\top Q \widetilde{P} \right) \left( I_n - 2\widetilde{I}_{p,n} \right) \right) \quad ((\text{Bendokat et al., 2024, Prop. 5.6})), \\
&= \frac{1}{2} \mathrm{mlog} \left( \widetilde{P}^\top (I_n - 2Q) \widetilde{P} \widetilde{P}^\top \left( I_n - 2\widetilde{P}\widetilde{I}_{p,n}\widetilde{P}^\top \right) \widetilde{P} \right) \\
&= \frac{1}{2} \mathrm{mlog} \left( \widetilde{P}^\top (I_n - 2Q) \left( I_n - 2\widetilde{P}\widetilde{I}_{p,n}\widetilde{P}^\top \right) \widetilde{P} \right), \\
&\overset{(1)}{=} \widetilde{P}^\top \left( \frac{1}{2} \mathrm{mlog} \left( (I_n - 2Q) \left( I_n - 2\widetilde{P}\widetilde{I}_{p,n}\widetilde{P}^\top \right) \right) \right) \widetilde{P}, \\
&\overset{(2)}{=} \widetilde{P}^\top \left( \frac{1}{2} \mathrm{mlog} \left( (I_n - 2Q)(I_n - 2P) \right) \right) \widetilde{P}, \\
&\overset{(3)}{=} \widetilde{P}^\top \overline{\Omega} \widetilde{P},
\end{aligned}
\tag{70}
$$

Eq. (70) follows from the following facts:

(1) When $\mathrm{mlog}(B)$ is well-defined and $A$ is non-singular, $A^{-1}(\log B)A = \log\left(A^{-1}BA\right)$ (Horn & Johnson, 2012).

(2) $P = \mathrm{mexp}([\overline{P}, \widetilde{I}_{p,n}])\widetilde{I}_{p,n} \, \mathrm{mexp}(-[\overline{P}, \widetilde{I}_{p,n}])$ (Nguyen, 2022a, Eq. (36)).

(3) Let $\overline{\Omega} = \frac{1}{2} \mathrm{mlog}\left((I_n - 2Q)(I_n - 2P)\right)$

Combining Eqs. (69) and (70), we have

$$
\begin{aligned}
\mathrm{d}_{\mathrm{gry}}(P, Q) &= \frac{1}{2} \left\| [\Omega, \widetilde{I}_{p,n}] \right\|, \\
&= \frac{1}{2} \left\| [\widetilde{P}^\top \overline{\Omega} \widetilde{P}, \widetilde{I}_{p,n}] \right\|, \\
&\overset{(1)}{=} \frac{1}{2} \left\| \widetilde{P}^\top [\overline{\Omega}, \widetilde{P}\widetilde{I}_{p,n}\widetilde{P}^\top] \widetilde{P} \right\|, \\
&\overset{(2)}{=} \frac{1}{2} \left\| [\overline{\Omega}, \widetilde{P}\widetilde{I}_{p,n}\widetilde{P}^\top] \right\|, \\
&\overset{(3)}{=} \frac{1}{2} \left\| [\overline{\Omega}, P] \right\|, \\
&\overset{(4)}{=} \frac{1}{2} \left\| \mathrm{Log}_P(Q) \right\|, \\
&\overset{(5)}{=} \left\| \mathrm{Log}_P(Q) \right\|_P, \\
&\overset{(6)}{=} \mathrm{d}_{\mathrm{geo}}(P, Q).
\end{aligned}
\tag{71}
$$

Eq. (71) follows from the following facts:

(1)
$$[\widetilde{P}^\top \overline{\Omega} \widetilde{P}, \widetilde{I}_{p,n}] = \widetilde{P}^\top \overline{\Omega} \widetilde{P} \widetilde{I}_{p,n} - \widetilde{I}_{p,n} \widetilde{P}^\top \overline{\Omega} \widetilde{P},$$
$$= \widetilde{P}^\top \left( \overline{\Omega} \widetilde{P} \widetilde{I}_{p,n} \widetilde{P}^\top - \widetilde{P} \widetilde{I}_{p,n} \widetilde{P}^\top \overline{\Omega} \right) \widetilde{P}, \tag{72}$$
$$= \widetilde{P}^\top [\overline{\Omega}, \widetilde{P} \widetilde{I}_{p,n} \widetilde{P}^\top] \widetilde{P}.$$

(2) Euclidean norm is invariant under the action $A \mapsto OAO^\top, \forall A \in \mathbb{R}^{n \times n}, O \in \mathrm{O}(n)$.

(3) $P = \mathrm{mexp}([\overline{P}, \widetilde{I}_{p,n}]) \widetilde{I}_{p,n} \mathrm{mexp}(-[\overline{P}, \widetilde{I}_{p,n}])$.

(4) $\mathrm{Log}_P(Q) = [\overline{\Omega}, P]$.

(5) $\|\cdot\|_P = \frac{1}{2} \|\cdot\|, \forall P \in \widetilde{\mathrm{Gr}}(p,n)$.

(6) $\mathrm{d}_{\mathrm{geo}}(P,Q) = \|\mathrm{Log}_P(Q)\|_P$ for any $P, Q \in \widetilde{\mathrm{Gr}}(p,n)$ not in each other's cut locus.

For $\{\mathrm{Gr}(p,n), \oplus^{\mathrm{Gr}}\}$, we first make the following notations: We denote the geodesic distance, gyrodistance, Riemannian logarithm at $I_{p,n}$, and Riemannian metric at $I_{p,n}$ on $\mathrm{Gr}(p,n)$ as $\mathrm{d}_{\mathrm{geo}}, \mathrm{d}_{\mathrm{gry}}$, $\mathrm{Log}_{I_{p,n}}$, and $\|\cdot\|_{I_{p,n}}^{\mathrm{ONB}}$. The counterparts on $\widetilde{\mathrm{Gr}}(p,n)$ are $\widetilde{\mathrm{d}_{\mathrm{geo}}}, \widetilde{\mathrm{d}_{\mathrm{gry}}}, \widetilde{\mathrm{Log}}_{\widetilde{I}_{p,n}}$, and $\|\cdot\|_{\widetilde{I}_{p,n}}^{\mathrm{PP}}$. As shown by Nguyen et al. (2024, App. N), $\pi : \mathrm{Gr}(p,n) \to \widetilde{\mathrm{Gr}}(p,n)$ is a Riemannian isometry. Then, for any $U, V \in \mathrm{Gr}(p,n)$ with $\pi(U) = P$ and $\pi(V) = Q$, we have the following:

$$\mathrm{d}_{\mathrm{gry}}(U,V) = \left\| \mathrm{Log}_{I_{p,n}} \left( \ominus^{\mathrm{Gr}} U \oplus^{\mathrm{Gr}} V \right) \right\|_{I_{p,n}}^{\mathrm{ONB}}$$
$$\overset{(1)}{=} \left\| \mathrm{Log}_{I_{p,n}} \left( \pi^{-1}(\widetilde{\ominus}^{\mathrm{Gr}} P \widetilde{\oplus}^{\mathrm{Gr}} Q) \right) \right\|_{I_{p,n}}^{\mathrm{ONB}}$$
$$\overset{(2)}{=} \left\| \pi_{*,I_{p,n}}^{-1} \left( \widetilde{\mathrm{Log}}_{\widetilde{I}_{p,n}} \left( \widetilde{\ominus}^{\mathrm{Gr}} P \widetilde{\oplus}^{\mathrm{Gr}} Q \right) \right) \right\|_{I_{p,n}}^{\mathrm{ONB}}$$
$$\overset{(3)}{=} \left\| \widetilde{\mathrm{Log}}_{\widetilde{I}_{p,n}} \left( \widetilde{\ominus}^{\mathrm{Gr}} P \widetilde{\oplus}^{\mathrm{Gr}} Q \right) \right\|_{\widetilde{I}_{p,n}}^{\mathrm{PP}} \tag{73}$$
$$= \widetilde{\mathrm{d}_{\mathrm{gry}}}(P,Q)$$
$$= \widetilde{\mathrm{d}_{\mathrm{geo}}}(P,Q)$$
$$\overset{(4)}{=} \mathrm{d}_{\mathrm{geo}}(U,V)$$

The above comes from the following.
(1) Gyro additions under the Riemannian isometry (Nguyen & Yang, 2023, Lem. 2.1).
(2,3,4) Riemannian operators under the Riemannian isometry (Gallier & Quaintance, 2020).

**Constant Curvature Spaces:**

When $\mathcal{M}_K = \mathbb{R}^n (K = 0)$, the gyro structures defined in Eq. (3)-Eq. (9) are reduced to the vector structures. The claim can be directly proved. In the following, we present the proof for $K \neq 0$. We first show the expression for the gyrodistance under $\mathcal{M}_K$, then the isometry of gyration, and finally the results on the gyroisometry of gyrotranslation and gyroinverse. In the following, $a, b, x, y$ are arbitrary points in $\mathcal{M}_K$.

*Gyrodistance and geodesic distance:* The Riemannian metric, logarithm, and geodesic distance on the CCS (Skopek et al., 2019) are

$$g_x^K = \left( \lambda_x^K \right)^2 g^{\mathrm{E}}, \tag{74}$$

$$\mathrm{Log}_x^K(y) = \frac{2}{\sqrt{|K|}\lambda_x^K} \tan_K^{-1} \left( \sqrt{|K|} \|-x \oplus_K y\| \right) \frac{-x \oplus_K y}{\|-x \oplus_K y\|}, \tag{75}$$

$$\mathrm{d}_{\mathrm{geo}}(x,y) = \frac{2}{\sqrt{|K|}} \tan_K^{-1} \left( \sqrt{|K|} \|-x \oplus y\| \right), \tag{76}$$

where $\lambda_x^K = \frac{2}{1+K\|x\|^2}$, $g^E$ is the standard Euclidean inner product, and $\oplus_K$ is the gyro addition in Tab. 2. Especially, when $x = 0$, we have

$$g_0^K = 2^2 g^E, \tag{77}$$

$$\mathrm{Log}_0^K(y) = \frac{1}{\sqrt{|K|}} \tan_K^{-1}\left(\sqrt{|K|}\,\|y\|\right) \frac{y}{\|y\|} \tag{78}$$

For the gyrodistance, we have the following:

$$\begin{aligned}
\mathrm{d}_{\mathrm{gry}}(x,y) &= \|-x \oplus y\|_{\mathrm{gr}}, \\
&= \|\mathrm{Log}_0(-x \oplus y)\|_0, \\
&= 2\left\| \frac{1}{\sqrt{|K|}} \tan_K^{-1}\left(\sqrt{|K|}\,\|-x\oplus y\|\right) \frac{-x\oplus y}{\|-x\oplus y\|} \right\|, \\
&= \frac{2}{\sqrt{|K|}} \tan_K^{-1}\left(\sqrt{|K|}\,\|-x\oplus y\|\right) \quad (\forall s > 0, \tan_K^{-1}(s) > 0),
\end{aligned} \tag{79}$$

where $\|\cdot\|_0$ is the norm induced by $g_0^K$.

*Norm invariance under gyration:* As $\mathcal{M}_K$ forms a real inner product gyrovector spaces (Ungar, 2009, Def. 3.2), any gyration on CCS preserves the norm induced by standard inner product:

$$\|\mathrm{gyr}[a,b](x)\| = \|x\|, \forall x. \tag{80}$$

For the gyronorm, we have the following:

$$\begin{aligned}
\|\mathrm{gyr}[a,b]x\|_{\mathrm{gr}} &= 2\|\mathrm{Log}_0(\mathrm{gyr}[a,b]x)\|_0, \\
&= \frac{2}{\sqrt{|K|}} \tan_K^{-1}\left(\sqrt{|K|}\,\|\mathrm{gyr}[a,b]x\|\right), \\
&= \frac{2}{\sqrt{|K|}} \tan_K^{-1}\left(\sqrt{|K|}\,\|x\|\right), \\
&= \|x\|_{\mathrm{gr}}.
\end{aligned} \tag{81}$$

*Isometry of left gyrotranslation and gyroinverse:* Note that $\mathcal{M}_K$ forms a gyrocommutative gyrogroup. According to Thm. 3.5, we can obtain the results. $\qquad\square$

### G.5 PROOF OF THM. 4.1

*Proof of Thm. 4.1.* According to Thm. 3.3 and Thm. 3.5, any left gyrotranslation is a gyroisometry. Therefore, for any $Q \in \mathcal{M}$, we have the following:

$$\begin{aligned}
\mathrm{d}_{\mathrm{gry}}(B \oplus P_i, Q) &\overset{(1)}{=} \mathrm{d}_{\mathrm{gry}}(\ominus B \oplus (B \oplus P_i), \ominus B \oplus Q) \\
&\overset{(2)}{=} \mathrm{d}_{\mathrm{gry}}((\ominus B \oplus B) \oplus \mathrm{gyr}[\ominus B, B](P_i), \ominus B \oplus Q) \\
&\overset{(3)}{=} \mathrm{d}_{\mathrm{gry}}(P_i, \ominus B \oplus Q).
\end{aligned} \tag{82}$$

The above comes from the following.

    (1) Any left gyrotranslation is a gyroisometry.

    (2) Left gyroassociative law.

    (3) $\ominus B \oplus B = E$ and pseudo-reduction.

Denoting the gyromean of $\{P_i\}$ and $\{B \oplus P_i\}$ as $M$ and $\widetilde{M}$, we have the following:

$$\begin{aligned}
B \oplus M &\overset{(1)}{=} B \oplus (\ominus B \oplus \widetilde{M}) \\
&\overset{(2)}{=} \mathrm{gyr}[B, \ominus B](\widetilde{M}) \\
&\overset{(3)}{=} \widetilde{M}.
\end{aligned} \tag{83}$$

The above comes from the following.

(1) Eq. (82) indicates that $M = \ominus B \oplus \widetilde{M}$.

(2) Left gyroassociative law.

(3) Pseudo-reduction.

Now, we proceed to deal with the second property. We have the following:

$$
\begin{aligned}
d_{\mathrm{gry}}(t \odot P_i, E) &\overset{(1)}{=} \|\ominus E \oplus (t \odot P_i)\|_{\mathrm{gr}} \\
&\overset{(2)}{=} \|t \odot P_i\|_{\mathrm{gr}} \\
&= \|t \operatorname{Log}_E P_i\|_E \\
&= |t| \|\operatorname{Log}_E P_i\|_E \\
&= |t| \|P_i\|_{\mathrm{gr}} \\
&\overset{(3)}{=} |t| \|\ominus E \oplus P_i\|_{\mathrm{gr}} \\
&= |t| d_{\mathrm{gry}}(E, P_i) \\
&\overset{(4)}{=} |t| d_{\mathrm{gry}}(P_i, E)
\end{aligned}
\tag{84}
$$

The above follows from the following.

(1) Symmetry of gyrodistance (Thm. 3.5).

(2) $\ominus E = E$.

(3) $P_i = \ominus E \oplus P_i$.

(4) Symmetry of gyrodistance.

The last equation in Eq. (84) indicates the homogeneity of dispersion from $E$. $\qquad\square$

### G.6 PROOF OF PROP. 5.1

*Proof of Prop. 5.1.* We only need to prove the equivalence of gyrodistance and geodesic distance. Note that every Lie group is automatically a gyrogroup with each gyration as the identity map.

We denote $\{\mathcal{M}, \oplus, g\}$ as a Lie group with left-invariant metric $g$. For any $P$ and $Q$ in $\mathcal{M}$, we have the following:

$$
\begin{aligned}
d_{\mathrm{gry}}(P, Q) &\overset{(1)}{=} \|\ominus P \oplus Q\|_{\mathrm{gr}} \\
&\overset{(2)}{=} \|\operatorname{Log}_E(\ominus P \oplus Q)\|_E \\
&= d_{\mathrm{geo}}(E, \ominus P \oplus Q) \\
&\overset{(3)}{=} d_{\mathrm{geo}}(P, P \oplus (\ominus P \oplus Q)) \\
&\overset{(4)}{=} d_{\mathrm{geo}}(P, Q).
\end{aligned}
\tag{85}
$$

The above derivation comes from the following.

1. Definition of gyrodistance Eq. (9).

2. Definition of gyronorm Eq. (8).

3. Under a left-invariant metric, any left Lie group translation is a Riemannian isometry.

4.
$$
\begin{aligned}
P \oplus (\ominus P \oplus Q) &= (P \ominus P) \oplus Q (\text{ the associative of group addition}) \\
&= E \oplus Q \\
&= Q.
\end{aligned}
\tag{86}
$$

Therefore, the gyromean and gyrovariance are exactly the Fréchet mean and variance under the geodesic distance, while the running mean updates are also identical under gyrodistance and geodesic distance. $\qquad\square$

### G.7 Proof of Prop. 6.1

We first review a fast and stable algorithm for the ONB Grassmannian logarithm (Bendokat et al., 2024, Alg. 5.3), and the calculation of Grassmannian logarithm under the projector perspective by the ONB Grassmannian logarithm (Nguyen et al., 2024, Prop. 3.12).

---

**Algorithm 2:** Grassmann logarithm under the ONB perspective (Bendokat et al., 2024, Alg. 5.3)

**Input:** $U, Y \in \mathrm{Gr}(p, n)$ are Stiefle representatives under ONB perspective.

1. $QSR^T \overset{\mathrm{SVD}}{:=} Y^T U$ with $S$ in ascending order, and $Q$ and $R$ column-wisely flipped accordingly;
2. $\hat{S} = \sqrt{I_n - S^2}$;
3. $\Delta = (I_n - UU^\top) Y Q \frac{\arcsin(\hat{S})}{\hat{S}} R^T$;

**Output:** $\mathrm{Log}_U(Y) = \Delta$

---

Alg. 2 reviews a fast and stable algorithm for the Grassmannian Riemannian logarithm under the ONB perspective $\mathrm{Gr}(p, n)$. The vanilla Riemannian logarithm in Tab. 8 requires an $n \times p$ SVD and a $p \times p$ matrix inverse, while Alg. 2 only requires an $p \times p$ SVD. Therefore, Alg. 2 is more efficient than the vanilla logarithm. Besides, Alg. 2 can also return a unique tangent vector when $Y$ is in the cut locus of $U$. For more details, please refer to Bendokat et al. (2024, Sec. 5.2).

As the projector perspective is isometric to the ONB perspective, the Grassmannian logarithm under the projector perspective can be calculated by the ONB Grassmannian logarithm (Nguyen et al., 2024, Prop. 3.12).

**Proposition G.2** ((Nguyen et al., 2024)). *Given any $P, Q \in \widetilde{\mathrm{Gr}}(p, n)$ with $U = \pi^{-1}(P)$ and $V = \pi^{-1}(Q)$, the Riemannian logarithm $\widetilde{\mathrm{Log}}_P(Q)$ on $\widetilde{\mathrm{Gr}}(p, n)$ is given as*

$$\widetilde{\mathrm{Log}}_P(Q) = \pi_{*,U}\left(\mathrm{Log}_U V\right), \tag{87}$$

*where* $\mathrm{Log}$ *is the Riemannian logarithm under the ONB perspective,* $\pi_{*,U} : T_U \mathrm{Gr}(p, n) \to T_P \widetilde{\mathrm{Gr}}(p, n)$ *is the differential map of $\pi$ at $U$, which is defined as*

$$\pi_{*,U}(\Delta) = \Delta U^\top + U \Delta^\top, \forall \Delta \in T_U \mathrm{Gr}(p, n). \tag{88}$$

Now, we begin to present the proof.

*Proof of Prop. 6.1.* We first show the expression for $\mathrm{Log}_{I_{p,n}}$ and $\widetilde{\mathrm{Log}}_{\widetilde{I}_{p,n}}$.

First note the following:

$$(I_n - I_{p,n} I_{p,n}^\top) = \begin{pmatrix} 0 & 0 \\ 0 & I_{n-p} \end{pmatrix}, \tag{89}$$

$$\begin{aligned} U^\top I_{p,n} &= (U_1^\top, U_2^\top) \begin{pmatrix} I_p \\ 0 \end{pmatrix} \\ &= U_1^\top, \end{aligned} \tag{90}$$

By the above two equations, the ONB Grassmannian logarithm at $I_{p,n}$ is

$$\begin{aligned} \mathrm{Log}_{I_{p,n}}(U) &= \begin{pmatrix} 0 & 0 \\ 0 & I_{n-p} \end{pmatrix} \begin{pmatrix} U_1 \\ U_2 \end{pmatrix} Q \frac{\arcsin(\hat{S})}{\hat{S}} R^T \text{ (Alg. 2)} \\ &= \begin{pmatrix} 0 \\ U_2 Q \frac{\arcsin(\hat{S})}{\hat{S}} R^T \end{pmatrix} \\ &= \begin{pmatrix} 0 \\ \widetilde{U}_2 \end{pmatrix}, \end{aligned} \tag{91}$$

where $QSR^T \overset{\mathrm{SVD}}{:=} U_1^\top$ with $S$ in ascending order, and $Q$ and $R$ column-wisely flipped accordingly, and $\hat{S} = \sqrt{I_n - S^2}$.

For $\widetilde{\mathrm{Log}}_{\widetilde{I}_{p,n}}$, we have

$$
\begin{aligned}
\widetilde{\mathrm{Log}}_{\widetilde{I}_{p,n}}(UU^\top) &\overset{(1)}{=} \pi_{*,I_{p,n}}\left(\mathrm{Log}_{I_{p,n}}(U)\right) \\
&\overset{(2)}{=} \pi_{*,I_{p,n}}\left(\begin{pmatrix} 0 \\ \widetilde{U}_2 \end{pmatrix}\right) \\
&\overset{(3)}{=} \begin{pmatrix} 0 & \widetilde{U}_2^\top \\ \widetilde{U}_2 & 0 \end{pmatrix}.
\end{aligned}
\tag{92}
$$

The above derivation comes from the following.

(1) Prop. G.2

(2) Eq. (91)

(3) For any $\Delta = (\Delta_1^\top, \Delta_2^\top)^\top \in T_{I_{p,n}}\mathrm{Gr}(p,n)$, where $\Delta_1$ is $p \times p$, we have the following

$$
\begin{aligned}
\pi_{*,I_{p,n}}\left(\begin{pmatrix} \Delta_1 \\ \Delta_2 \end{pmatrix}\right) &= \begin{pmatrix} \Delta_1 \\ \Delta_2 \end{pmatrix}(I_p, 0) + \begin{pmatrix} I_p \\ 0 \end{pmatrix}(\Delta_1^\top, \Delta_2^\top) \\
&= \begin{pmatrix} \Delta_1 & 0 \\ \Delta_2 & 0 \end{pmatrix} + \begin{pmatrix} \Delta_1^\top & \Delta_2^\top \\ 0 & 0 \end{pmatrix} \\
&= \begin{pmatrix} \Delta_1 + \Delta_1^\top & \Delta_2^\top \\ \Delta_2 & 0 \end{pmatrix}.
\end{aligned}
\tag{93}
$$

Combining all the above results together, we have the following:

$$
\begin{aligned}
[\overline{UU^\top}, \widetilde{I}_{p,n}] &= \left[\widetilde{\mathrm{Log}}_{\widetilde{I}_{p,n}}(UU^\top), \widetilde{I}_{p,n}\right] \\
&= \left[\begin{pmatrix} 0 & \widetilde{U}_2^\top \\ \widetilde{U}_2 & 0 \end{pmatrix}, \widetilde{I}_{p,n}\right] \\
&= \begin{pmatrix} 0 & \widetilde{U}_2^\top \\ \widetilde{U}_2 & 0 \end{pmatrix}\begin{pmatrix} I_p & 0 \\ 0 & 0 \end{pmatrix} - \begin{pmatrix} I_p & 0 \\ 0 & 0 \end{pmatrix}\begin{pmatrix} 0 & \widetilde{U}_2^\top \\ \widetilde{U}_2 & 0 \end{pmatrix} \\
&= \begin{pmatrix} 0 & -\widetilde{U}_2^T \\ \widetilde{U}_2^T & 0 \end{pmatrix}.
\end{aligned}
\tag{94}
$$

$\square$

