# OpenReview forum: "Gyrogroup Batch Normalization"
_ICLR.cc/2025/Conference — ICLR 2025 Poster_

### Official Review · Reviewer_Yeux · 2024-10-31

**Soundness:** 3
**Presentation:** 4
**Contribution:** 3
**Rating:** 8
**Confidence:** 2

**Summary:**

The paper proposes a new Riemannian Batch Normalization (RBN), named Gyrogroup Batch Normalization. This new RBN method provides controllable means and variances and includes several previous RBN methods. To include the case of Grassmannian, this work proposes the pseudo-reductive gyrogroups that lie between the gyrogroup and the nonreductive one. Theoretical analysis regarding this new approach is provided with numerical verifications across several experiments.

**Strengths:**

The paper is well-written and well-structured and the comparisons with previous RBN methods are comprehensive. In particular, the comparison of assumptions is presented clearly. The theories seem solid and the performance is improved compared to the existing methods.

**Weaknesses:**

I suggest the authors provide a comprehensive version of Algorithm 1 to specify the calculation of each quantity or elaborate in the appendix.

**Questions:**

1. What is the computational complexity of GyroBN, and how does it scale with the input dimension compared to previous methods?

2. How do you predetermine the running mean, running variance, biasing parameter, and scaling parameter in Algorithm 1?

3. Can you provide any discussion if the gyrations are not isometric?

---

> ### Author Response · Authors · 2024-11-19
> **Response to Reviewer Yeux**
>
> We thank Reviewer $\textcolor{red}{Yeux}$​ for the constructive suggestions and insightful comments! In the following, we respond to the concerns in detail. 😄
> ***
>
> # 1. Summary tables of Grassmannian and hyperbolic GyroBNs.
>
> The specific implementation of Alg. 1 on a gyrogroup can be carried out in a plug-in manner by substituting the required operators from Tab. 2 and Tab. 8 into Alg. 1. This approach is how we construct the specific Grassmannian and hyperbolic GyroBNs detailed in Sec. 6. To improve readability and accessibility, we have added a summary table in App. G, which comprehensively lists all the required operators for the Grassmannian and hyperbolic GyroBNs.
> ***
>
> # 2. Computational efficiency: Our GyroBN is more efficient than previous Riemannian BN methods.
>
> We compared the training efficiency of our GyroBN with previous BN methods on both Grassmannian and hyperbolic spaces. On the most representative dataset for the Grassmannian, the training time per epoch for GyroBN is approximately **one-tenth** of that required by previous BN methods. For further details, please refer to CQ#1 in the common response.
> ***
>
> # 3. Initialization in Alg. 1.
>
> The initialization generally aligns with the standard Euclidean BN [a]. However, on the hyperbolic space, we follow [b, Alg. 1] to initialize the running mean and variance for a fair comparison.
>
> - **Running mean and variance:** Similar to the standard Euclidean BN, they are initialized as the identity element ($I_{p,n}$) and 1 on the Grassmannian. On the hyperbolic space, following [b, Alg. 1], we initialize the running mean and variance as the first batch mean and variance.
> - **Biasing parameter:** Similar to the standard Euclidean BN, it is initialized as the identity element: $I_{p,n}$ for the Grassmannian and the zero vector for the hyperbolic space.
> - **Scaling parameter:** Similar to the standard Euclidean BN, it is initialized as 1.
> ***
>
> # 4. Discussion on gyrogroups with non-isometric gyrations.
>
> **Hypothesis A:** Any gyration over the gyrogroup $G$ is a gyroisometry.
>
> If Hyp. A does not hold, GyroBN can still be applied numerically; however, its ability to control sample statistics is not guaranteed.
>
> Recalling Eq. (59) in the proof of Thm. 3.5, we observe that the gyroisometry of gyrations is the necessary and sufficient condition for the gyro left translation to be a gyroisometry (by Thm. 3.3). Combined with the proof of Thm. 4.1, this implies that GyroBN can normalize the sample statistics only if all gyrations in the gyrogroup are gyroisometries.
>
> That said, as demonstrated in Prop. 3.6, Hyp. A holds for all gyrogroups currently used in machine learning, ensuring that GyroBN maintains its theoretical guarantees in these cases. However, we acknowledge that in the future, gyrogroups with non-isometric gyrations might be introduced. This remains an open question and could present interesting challenges for extending the current framework of GyroBN.
> ***
>
> # References
> > [a] Batch normalization: Accelerating deep network training by reducing internal covariate shift
> >
> > [b] Differentiating through the fréchet mean

---

> > ### Comment · Reviewer_Yeux · 2024-11-24
> >
> > Thank you for the detailed response and clarification. I would like to keep my rating.

---

> > > ### Author Response · Authors · 2024-11-24
> > >
> > > Thanks for the reply! We are grateful for your time and efforts. 😄

---

### Official Review · Reviewer_A3ej · 2024-11-01

**Soundness:** 3
**Presentation:** 3
**Contribution:** 2
**Rating:** 8
**Confidence:** 4

**Summary:**

This paper proposes a geometric neural network batch normalization operation that is based off of a proposed gyrodistance frechet mean.

**Strengths:**

1. The paper's method seems principled, and the derivations seem right to me.
2. The empirical results are reasonably convincing, showing good improvement over prior methods.

**Weaknesses:**

Overall, there are no major concerns from me. I believe that the paper is technically sound and complete, but the scope is rather limited and the overall contribution may ultimately be incremental (over prior Riemannian batch norm papers). Beyond that, I have two minor nitpicks:

1. A natural issue is that not all manifolds are gyrogroups. This means that the proposed method is not general. This is probably okay since most manifold methods are not general (by dint of the fact that it is hard to work on manifolds), but might affect some use cases that were not covered.
2. The empirical results are not very convincing. In particular, while they are controlled against other RBN methods, the base architecture is often weak/basic. I would like to see comparisons with other architectures (e.g. HNN++, Riemannian ResNet [1]). This may also be important since the architectures used are built solely from gyro-operations, and using different architectures could control for this confounding factor (gyro operations helping gyro operations).

[1] https://arxiv.org/abs/2310.10013

**Questions:**

N/A

---

> ### Author Response · Authors · 2024-11-19
> **Response to Reviewer A3ej**
>
> We thank Reviewer $\textcolor{blue}{A3ej}$ for the thoughtful comments. We respond to the concerns as follows.😄
> ***
>
> # 1. The Applicability of GyroBN.
> We have discussed the limitation of GyroBN regarding its inapplicability to non-gyrogroup manifolds at the beginning of the appendix. However, GyroBN already supports seven non-trivial gyrogroups, as shown in Tab. 2, as well as all trivial gyrogroups, including various matrix Lie groups and Euclidean space. Furthermore, several previous Riemannian batch normalization methods are special cases of GyroBN, as discussed in Sec. 5. We hope that our work will advance the development of deep networks for data with non-trivial geometries.
> ***
>
> # 2. Additional experiments: HNN++ [a] and RResNet [b] could benefit from our GyroBN.
> Following the reviewer's constructive suggestion, we further conduct experiments on the two mentioned hyperbolic baselines, HNN++ and RResNet. We use the official code to reimplement the experiment.
>
> ## Experiments on HNN++
>
> We evaluate HNN++ with and without RBN-H [c] and our GyroBN-H on the Cora, Disease, and Airport datasets for the link prediction task. Like the HNN experiments, we used a two-layer HNN++ backbone as the encoder, with hyperbolic hidden feature dimensions set to 128. Tab. A reports the five-fold average results, leading to the following findings:
>
> - **GyroBN can improve and stabilize HNN++.** While RBN-H could degrade HNN++’s performance, particularly on the Disease dataset, GyroBN consistently improves it. Additionally, we observed that HNN++ could be sensitive to weight decay. For instance, on the Cora dataset, setting the weight decay to $1e^{-3}$ reduces the performance to 62.55, prompting us to use zero weight decay for HNN++ on this dataset. In contrast, HNN++ with GyroBN is robust to weight decay. Furthermore, HNN++-GyroBN achieves the smallest performance variance across all three datasets, indicating that GyroBN facilitates more robust learning.
> - **GyroBN Accelerates Convergence.** On the Cora dataset, HNN++ requires over 400 epochs to converge. In contrast, HNN++-GyroBN converges in approximately 150 epochs, significantly improving convergence.
>
> Table A: Comparison of HNN++ with or without RBN-H and GyroBN-H on the Cora, Disease, and Airport datasets.
> |     Dataset    |     Cora     |    Disease    |    Airport   |
> |:--------------:|:------------:|:-------------:|:------------:|
> |      HNN++     | 91.06 ± 0.47 |  77.83 ± 1.39 | 94.93 ± 0.23 |
> |   HNN++-RBN-H  | 88.22 ± 0.67 |  59.50 ± 1.67 | 94.91 ± 0.23 |
> | HNN++-GyroBN-H | **92.83 ± 0.32** |  **79.53 ± 0.74** | **95.89 ± 0.10** |
>
> ## Experiments on RResNet
>
> We focus on the RResNet Horo [b], which leverages a vector field induced by the horosphere projection feature map. Following the settings of HNN and HNN++, we use 2 residual blocks as the backbone. For a more comprehensive comparison, we evaluate hidden feature dimensions ranging from 8 to 128. Tab. B reports the five-fold average results on the Cora dataset, leading to the following findings:
>
> **GyroBN consistently improves performance.** While RBN performs well only under the 128-dimensional setting, it fails under other dimensions, performing worse than the baseline RResNet. In contrast, GyroBN consistently improves RResNet’s performance across all hidden feature dimensions.
>
> **GyroBN stabilizes performance and preserves representation power.** Both RResNet and RResNet-RBN exhibit relatively big fluctuations in performance across different hidden dimensions. However, RResNet-GyroBN shows a more stable performance. This indicates that our GyroBN could maintain the representation power of the backbone network.
>
> Table A: Comparison of RResNet with or without RBN-H and GyroBN-H under different hidden feature dimensions on the Cora dataset.
> |        Dim       |       8       |       16       |       32      |      64     |     128    |
> |:----------------:|:-------------:|:--------------:|:-------------:|:-----------:|:----------:|
> |      RResNet     |  71.77 ± 6.67 |  77.38 ± 10.82 |  77.14 ± 7.48 | 66.73±11.85 | 80.75±4.12 |
> |   RResNet-RBN-H  |  61.94 ± 2.28 |   63.54 ± 3.01 |  62.26 ± 3.48 |  60.38±2.97 | 87.92±2.67 |
> | RResNet-GyroBN-H |  **84.58 ± 5.44** |   **87.67 ± 2.53** |  **88.08 ± 2.90** |  **86.50±1.59** | **89.52±3.42** |
>
> *Remark: The above discussion has been added to the revised submission (App. F. 6.)*
> ***
>
> # References
> > [a] Hyperbolic neural networks++
> >
> > [b] Riemannian residual neural networks
> >
> > [c] Differentiating through the fréchet mean

---

> ### Comment · Reviewer_A3ej · 2024-11-26
> **Thank you for your response**
>
> Thank you for your response. The experimental results seem rather promising.
>
> Overall, I'll raise my score to an 8. I would be more comfortable with a 7 (a weak accept versus 8 accept, 6 marginal accept), but that does not seem to be an option.

---

> > ### Author Response · Authors · 2024-11-26
> >
> > Thanks for the reply and careful review!  😄

---

### Official Review · Reviewer_MrZQ · 2024-11-02

**Soundness:** 3
**Presentation:** 2
**Contribution:** 3
**Rating:** 6
**Confidence:** 2

**Summary:**

This paper addresses the problem of extending batch normalization, which has accelerated deep learning in many areas, to non-Euclidean geometries, where the data lies on a Riemannian manifold. While computing mean and variance are trivial in Euclidean spaces, they become non-trivial in general Riemannian manifolds, because there is no analogue of addition that is associative, and would allow for an easy computation of the mean. The main goal is to extend the algebra for Riemannian manifolds, where mean and variance in the BN operator should be replaced by  Fréchet mean and Fréchet variance over the manifold. The main theoretical contribution of the work is unifying the analysis of some previous works that cover special cases of Riemannian manifolds, and extending them to some new types of Riemannian manifolds, so that batch normalization can be defined and computed. They go on to show the empirical value of their proposed method by testing newly developed Grassmannian network, GyroGr, with different types of batch normalization, and their proposed normalization improves the test accuracies.

The paper builds on previous works that has addressed these problems for specific forms of Riemannian manifolds (Table 1). However, authors argue that a more general approach can combine these as special cases, thereby unifying the analysis. Furthermore, they introduce a more relaxed notion than earlier works, termed "pseudo-reductive gyrogroup" that encompasses previous works, but can admit new Riemannian structures due to its relaxed condition, such as Grassmannian manifolds. More specifically, they introduce the relaxed condition, pseudo-reductive, which replaces left reduction (G4) condition that is more stringent, which covers Grassmannian manifolds.

**Update**
In light of the other discussions as well as the explanations and clarifications by the authors, I'm happy to increase my score from 5 (marginally below acceptance) to 6 (marginally above the acceptance)

**Strengths:**

Normalization layers and in particular batch normalization have been key to success in training many modern architectures. However, when physical or mathematical constraints imply non-Euclidean geometries where data is on a manifold, computing mean and variance for BN becomes a non-trivial task. While one could take an ad hoc or heuristic approach, a more principled approach is likely to further our understanding and improve the state of the art in this field.

This paper addresses this problem by studying a  relaxed set of properties, that admit a sufficiently rich algebra for computing mean and variance, and thereby a BN operator. This principled approach is certainly a welcome development.  To the extent that I could understand which was admittedly limited, the theoretical statements and assertions seem to be sound. Finally, in section 7 they present experiments mostly on GyroGr which is a Grassmannian network as a backbone, and compare it without BN, or with other types of BN, and show their proposed BN achieves the best results. While the empirical setup is sound, I cannot assess the significance of these results, given that they mostly focus on one network and one particular type of Riemannian manifold.

**Weaknesses:**

While this work seems to be on solid theoretical grounds and is mathematically sound, my main concern is the overall structure and writing, which make the theoretical contributions of the paper rather hard to grasp. I've added a few concrete suggestions that I believe would improve the readability of this paper.

- An expanded preliminary on various concepts non-Euclidean geometry. While authors can assume that their readers is limited to those with deep and expert knowledge of this topic, such an assumption would narrow  the audience of this paper quite a lot. For reference, I spent >5 hours to try educate myself on some of the basics that I did not know before reading the paper, but besides "parsing" the theoretical statements, I could not gain a deeper insight. With an expanded preliminary that illuminates these concepts, I could have understood the results on a much deeper level.
- Adding high level ideas : For theoretical contributions, it is always valuable to the reader to get a high level view of the ideas, techniques, or proof insights. In this particular case of Riemannian geometries, which presents itself as highly abstract, these types of explanations are even more crucial.
- Adding concrete examples for abstract concepts: there are numerous technical terms, some classical and some introduced in the paper, that are left without any high level explanations, eg. gyrations, left gyroassociative law, left reduction law, pseudo-reductive gyrogroups, are all concepts that are formally introduced, but no high level explanations are given. So the reader is forced to just parse and compile the formal statements, which is not easy, unless they are already an expert on this topic. Would it be possible for authors to give concrete examples for each concept as they go along with their formal definitions and statements? For example, can they use a working example of a Grassmannian, and explain the rough geometric intuition behind each notion?  While that might not be easy in all cases, it would tremendously improve the readability and value of the paper.

**Questions:**

No questions.

---

> ### Author Response · Authors · 2024-11-19
> **Response to Reviewer MrZQ**
>
> We thank Reviewer $\textcolor{green}{MrZQ}$ for the careful review and the suggestive comments. Below, we address the comments in detail.😄
> ***
>
> # 1. GyroBN on other hyperbolic baselines.
>
> We have conducted additional experiments during the rebuttal, including evaluations of GyroBN on two hyperbolic baselines: HNN++ [a] and Riemannian ResNet (RResNet) [b]. The results demonstrate that GyroBN improves and stabilizes the backbone networks while accelerating convergence. For more details, please refer to the **2nd response to $\textcolor{blue}{A3ej}$** or **App. F.6**.
> ***
>
> # 2. Expanded preliminaries.
>
> We thank the reviewer for their constructive comments. In the revised submission, we have expanded the preliminaries by adding a subsection to introduce the basics of Riemannian geometry. To avoid the burden of symbols, we explain Riemannian operators—such as geodesics, the Riemannian log, the Riemannian exp, and parallel transport—through their geometric reinterpretations of Euclidean vector operations. Additionally, we also recap Lie groups and include several simple examples for better understanding.
> ***
>
> # 3. Adding high-level ideas into the introduction.
>
> The key operations in Euclidean BN consist of centering (vector subtraction), biasing (vector addition), and scaling (vector scalar product). Since gyro spaces are natural extensions of Euclidean spaces, we use gyro subtraction, addition, and scalar product to construct GyroBN. To conduct an in-depth theoretical analysis, we introduce the relaxed pseudo-reductive gyrogroups. To reflect this high-level idea, we have revised the introduction and added the following into **L 53** in the revised version:
>
> - *We use the gyro addition, subtraction, and scalar product to extend the centering (vector subtraction), biasing (vector addition), and scaling (vector scalar product) in the Euclidean BN into manifolds in a principled manner. For broader applicability and in-depth theoretical analysis, we adapt the existing gyrogroup into a more relaxed structure, termed pseudo-reductive gyrogroup.*
> ***
>
> # 4. Intuitive explanations of abstract concepts.
>
> We thank the reviewer for their constructive suggestion. We agree that some concepts in gyrogroups can appear abstract. The best way to understand them is to contrast them with the trivial case of $\mathbb{R}$. In the revised submission, we have added intuitive explanations for several key concepts, including gyrations, the left gyroassociative, the left reduction, and pseudo-reduction, into **App. C.4**.
>
> Following your suggestion, we also use the Grassmannian as an example to illustrate some concepts. However, due to the cumbersome concrete expressions of the Grassmannian gyro operations, we use notations, such as $\oplus^{\mathrm{Gr}}$, to maintain clarity while still providing the necessary intuition. Additionally, all the involved Riemannian and gyro structures across different manifolds are summarized in **Tabs. 2 and 8**, offering a comprehensive overview. We hope these make the abstract concepts more accessible to readers.
>
> ***
> # References
> > [a] Hyperbolic neural networks++
> >
> > [b] Riemannian residual neural networks

---

> > ### Comment · Reviewer_MrZQ · 2024-11-22
> >
> > I thank the authors for the further clarifications. I highly welcome the addition of the new sections explaining the basics. I had a look at the added sections and they clarify a lot of important details that I found hard to understand the first time. I have no further concerns about the paper.

---

> > > ### Author Response · Authors · 2024-11-22
> > > **Response to Reviewer MrZQ**
> > >
> > > Thanks for the reply! We appreciate your time and efforts during the review and discussion. 😄

---

### Official Review · Reviewer_oCvr · 2024-11-03

**Soundness:** 3
**Presentation:** 3
**Contribution:** 3
**Rating:** 6
**Confidence:** 3

**Summary:**

The paper proposes a new batch normalization method on Riemannian manifolds named GyroBN. GryoBN is a new batch algorithm on a class of algebras geometries named “pseudo-reductive gyrogroups”, which relaxes the left reduction law of a gryogroup as defined by [Ungar 2009]. The work generalizes prior work on batch normalization on Lie groups, such as LieBN [Chen et al., 2024b], and the SPD batch norm proposed by Brooks et al., 2019.

**Strengths:**

To me, the main contribution of the work is how it theoretically generalizes several prior work on batch norm for manifolds. The work generalizes LieBN [Chen et al., 2024b] and the SPD Batch norm proposed in [Brooks et al., 2019] to a wider class of geometries. While there have been some work such as [Lou et al., 2020] which propose general batch norm algorithms, it does not give control over sample statistics. Although the originality of extending prior work on gyrononreductive gyrogroups (as done by [Nguyen, 2022] to the proposed pseudo-reductive gryogroups) seems limited, it still appears to be an important extension.

**Weaknesses:**

While the paper provides experiments on Grasmannian and hyperbolic manifolds, it is not clear from the paper why empirically GryroBN outperforms prior work. Is there a theoretical reason why using gryodistance may outperform prior work that use geodesic distance for batch norm? I wish the paper would do a deeper analysis into this difference, and how it affects what the networks are learning. Here are some possible experiments that could strengthen the paper's analysis:
- The original reason behind batch norm for Euclidean neural networks is covariate shift. Can you show empirically that the baselines suffer from covariate shift, and that GryoBN reduces covariate shift? Even a result showing the opposite would be an interesting contribution to the geo DL community.
- Another experiment is to analyze the condition numbers of network's layers. Is is true that GryoBN makes the outputs of the neural networks better conditioned?



Another weakness is that the paper does not analyze how efficient the proposed batch norm algorithm is compared to prior work. Prior work based on Karcher flow are known to be slow. It would strengthen the paper to include this analysis for the experiments on Grassmannian and hyperbolic spaces. I see that there is a table showing training efficiency in Appendix F.1.4, but it only a comparison to prior work without batch normalization, not other work that do use batch norm.

**Questions:**

- Is there a reason why the notations for the identity and left inverse different in Eq. 12 than the ones used in Section 2? Is Eq 12 equivalent to saying that $gry[ \ominus b, b] = e$?
- How efficient is the mean and variance calculation? Prior work based on Karcher flow are known to be slow. How does GyroBN’s speed compare to prior work?

---

> ### Author Response · Authors · 2024-11-19
> **Response to Reviewer oCvr (Part 1)**
>
> We thank Reviewer $\textcolor{purple}{oCvr}$ for the encouraging feedback and the constructive comments! In the following, we respond to the concerns point by point. 😄
> ***
>
> # 1. The empirical superiority of GyroBN stems from its unique ability to control sample statistics.
>
> Controlling sample statistics is a crucial factor behind the success of Euclidean batch normalization, a capability absent in several previous Riemannian batch normalization methods. In contrast, our proposed GyroBN effectively extends this functionality to the gyro space, as established in Thm. 4.1.
> ***
>
> # 2. Gyrodistance is identical to the geodesic distance within all gyrogroups in Tab. 2, commonly used in machine learning.
>
> While there is no general guarantee that gyrodistance is identical to geodesic distance across all gyrogroups, our Prop. 3.6 demonstrates that the two distances are identical for the gyrogroups listed in Tab. 2. These include three SPD, two Grassmannian, and two constant curvature space gyrogroups, all of which are frequently encountered in machine learning applications. For these gyrogroups, the Fréchet mean, Fréchet variance, and running mean update under the gyrodistance are equivalent to their geodesic counterparts.
>
> Furthermore, if there will be a gyrogroup in machine learning, whose gyrodistance is different from its geodesic ones, we could also expect the rationality of gyrodistance. As the data distribution in the GyroBN (Fréchet mean and variance) are defined by the gyrodistance, our GyroBN can still normalize the data distribution in the sense of gyro calculus. As gyro operations intuitively help gyro networks (mentioned by reviewer A3ej), we also expect our GyroBN can be a compelling algorithm. However, at present, we have not encountered such gyrogroups in machine learning.
> ***
>
> # 3. Numerical experiments show that GyroGr and HNN could bring covariate shifts.
>
> Due to the complexity of the manifold, Riemannian networks usually do not maintain the data distribution, highlighting the utility of our GyroBN. To better analyze covariate shifts, we conduct numerical experiments over the Grassmannian and hyperbolic networks.
>
> - **Grassmannian**: The transformation and pooling layers can be expressed as
> $$f _{trans}: \mathrm{Gr}(p,n) \rightarrow \mathrm{Gr}(p,n),$$
> $$f _{pooling}: \mathrm{Gr}(p,n) \rightarrow \mathrm{Gr}(p,n/2),$$
> (assuming $n$ is even).
>
>   Since the ONB Grassmannian $\mathrm{Gr}(p, n)$ is a quotient manifold and pooling changes dimensions, we use the geodesic distance between the batch mean and the identity element $I _{p, n} = (I _p, \mathbf{0})^\top \in \mathbb{R}^{n \times p}$ as a consistent measure. Note that the geodesic distance on $\mathrm{Gr}(p, n)$ is bounded by $\sqrt{p}\frac{\pi}{2}$ [a, Thm. 8].
>
>   We generated 30 random Grassmannian matrices of size $100 \times 10$ with an initial batch mean as the identity element $I _{p, n}$. We denote the resulting batch mean after transformation or pooling as $M _{out}$. If the distribution were preserved, the geodesic distance $d(M _{out}, I _{p, n})$ (or $d(M _{out}, I _{p, n/2})$ for pooling) would be zero. Tab. A shows that $M _{out}$ significantly deviates from $I _{p, n}$, indicating a covariate shift.
>
> Table A: Ten-fold results for geodesic distance $d(M _{out}, I _{p, n})$ and shift $\Delta = \frac{d(M _{out}, I _{p, n})}{b} \times 100$, where $b = \sqrt{p} \frac{\pi}{2}$.
>
> |             |  Pooling       |  Transformation     |
> |:------------:|:--------------------:|:---------------------:|
> | $d(M _{out}, I _{p, n})$  | $2.90 \pm 0.11$   | $4.18 \pm 0.04$      |
> | $\Delta$   \( \% \)      | $58.47 \pm 2.28$ | $84.22 \pm 0.72 $ |
>
> - **Hyperbolic Space**: We examine covariate shift in the transformation layer (Möbius matrix-vector multiplication [b, Lem. 6]) in HNN. We focus on the canonical Poincaré ball (curvature $K=-1$). We randomly generate 30 5-dimensional Poincaré vectors. The batch mean vectors of input and output in the transformation layer are:
> $$
> M_{in} = [-0.0055,  0.0503, -0.0913, -0.0493,  0.0652]^\top
> $$
> $$
> M_{out} = [-0.0969, -0.0443,  0.0385,  0.0936, -0.0770]^\top
> $$
>   The geodesic distance between $M_{in}$ and $M_{out}$ is $0.55$, indicating the covariate shift in HNN. We agree that analyzing the covariate shift is helpful to understand the impact of BN. We will add the above discussion to the supplementary.
> ***

---

> ### Author Response · Authors · 2024-11-19
> **Response to Reviewer oCvr (Part 2)**
>
> # 4. GyroBN effectively reduces condition numbers of weight and network Jacobian matrices
>
> In the GyroGr network, the transformation layer outputs $n \times p$ Grassmannian matrices, whose condition numbers are trivial since all singular values of a Grassmannian matrix are 1. Similarly, for the HNN model, the transformation layer outputs a vector, leading to a trivial condition number as well. Thus, we focus on evaluating the condition numbers of the weight matrices in the hidden transformation layer and the overall network Jacobian matrix. Without loss of generality, we conduct evaluations on Grassmannian GyroGr networks under one block architecture, both with and without different BN layers, using each trained model. The results demonstrate that GyroBN consistently reduces the condition numbers of the weight matrices and the network Jacobian.
>
> ## Condition numbers of weight matrices in the GyroGr transformation layer
>
> Table B: Statistics of condition numbers for the weight matrices in the GyroGr transformation layer across different normalization methods. As weight matrices have 8 channels, we report their mean, standard deviation (std), minimum, and maximum values. The dimensions of the 8-channel transformation matrices are $83 \times 10$ on the HDM05 dataset and $140 \times 10$ on the other datasets.
>
> | Dataset | BN               | Mean ± Std   | Min   | Max   |
> |---------|------------------|--------------|-------|-------|
> | HDM05   | None             | 3.81 ± 0.23  | 3.46  | 4.17  |
> |         | ManifoldNorm-Gr  | 1.97 ± 0.15  | 1.83  | 2.23  |
> |         | RBN-Gr           | 3.49 ± 0.46  | 2.96  | 4.43  |
> |         | GyroBN-Gr        | 2.37 ± 0.18  | 2.12  | 2.67  |
> | NTU60   | None             | 2.77 ± 0.12  | 2.61  | 2.94  |
> |         | ManifoldNorm-Gr  | 1.76 ± 0.10  | 1.63  | 1.91  |
> |         | RBN-Gr           | 2.17 ± 0.24  | 1.94  | 2.51  |
> |         | GyroBN-Gr        | 2.02 ± 0.14  | 1.82  | 2.27  |
> | NTU120   | None             | 3.35 ± 0.28  | 2.97  | 3.72  |
> |         | ManifoldNorm-Gr  | 1.91 ± 0.10  | 1.80  | 2.14  |
> |         | RBN-Gr           | 2.22 ± 0.11  | 2.10  | 2.36  |
> |         | GyroBN-Gr        | 2.16 ± 0.11  | 2.00  | 2.33  |
>
> As shown in Tab. B, GyroBN consistently reduces the condition numbers of the weight matrices across datasets. Interestingly, ManifoldNorm achieves the lowest condition numbers. However, its performance is even worse than the GyroGr baseline, suggesting that excessively reducing the condition number might limit the model’s capacity to capture sufficient expressive features.
>
> ## Condition number of network Jacobian (output w.r.t. input)
>
> Table C: Condition numbers of network jacobian (output w.r.t. input) across different normalization methods. The statistics come from 100 random samples. The dimensions of Jacobian matrices are $117 \times 930$ on the HDM05 dataset and $11 \times 9600$ on the NTU120 dataset.
> | Dataset  | Method              | Mean ± Std       | Min     | Max     |
> |----------|---------------------|------------|---------|---------|
> | HDM05    | None               | 61.05 ± 4.15 | 52.19   | 74.29   |
> |          | ManifoldNorm-Gr    | 67.38 ± 6.82 | 51.15   | 88.81   |
> |          | RBN-Gr             | 64.16 ± 6.44 | 50.54   | 80.07   |
> |          | GyroBN-Gr          | 59.85 ± 4.33 | 48.46   | 70.47   |
> | NTU120   | None               | 122.24 ± 68.45 | 61.8    | 399.28  |
> |          | ManifoldNorm-Gr    | 116.75 ± 65.17 | 75.46   | 443.01  |
> |          | RBN-Gr             | 108.75 ± 52.27 | 63.19   | 349.25  |
> |          | GyroBN-Gr          | 97.01 ± 41.88  | 61.19   | 262.55  |
>
> We further analyze the condition number of the network Jacobian on the HDM05 and NTU120 datasets. We randomly select 100 samples to calculate the statistics of the associated network condition numbers, as shown in Tab. C. Our GyroBN consistently lowers condition numbers compared to other methods across both datasets, indicating that GyroBN stabilizes network training. Additionally, the reduced maximum condition numbers suggest that GyroBN avoids extreme outliers in conditioning, further highlighting its advantage over prior methods. Interestingly, on the HDM05 dataset, both ManifoldNorm and RBN increase the network conditioning.
> ***

---

> > ### Author Response · Authors · 2024-11-19
> > **Response to Reviewer oCvr (Part 3)**
> >
> > # 5. Our GyroBN is more efficient than other RBN methods.
> >
> > We compare GryoBN against other BN methods on the Grassmannian and hyperbolic spaces. Our GyroBN is more efficient, especially on the Grassmannian manifold. On the most representative dataset for the Grassmannian, the training time per epoch for GyroBN is approximately **one-tenth** of that required by previous BN methods. Please refer to the common response **CQ#1** for the detailed response.
> > ***
> >
> > # 6. Explanation of Eq. (12).
> >
> > Let us recall Eq. (12):
> > $$\operatorname{gyr}[X, P] = \mathbb{1}, \text{ for any left inverse } X \text{ of } P \text{ in } G.$$
> >
> > **$\ominus P$ and $X$.** Here, $X$ can be intuitively understood as $\ominus P$, as described in Sec. 2. However, the notation $\ominus P$ can only be used when the uniqueness of the left inverse is established. Up to Eq. (12), as we have relaxed the definition of gyrogroups, the uniqueness of the left inverse is not yet guaranteed. Later, in Thm. D.1 we prove the uniqueness of the left inverse. For the sake of mathematical rigor, we used "any left inverse $X$ of $P$" in Eq. (12) rather than $\ominus P$. Mathematical rigor can sometimes feel like both a curse and a blessing—it comes with the burden of handling intricate symbols but ensures clarity and consistency. :-)
> >
> > **$\mathbb{1}$ is the identity map, not the identity element.** Any gyration is a map (automorphism): $\operatorname{gyr}[A, B]: G \rightarrow G$ for any $A$ and $B$ in the gyrogroup $G$. Thus, $\mathbb{1}$ refers to the identity map:
> > $$
> > \mathbb{1}(A) = A, \quad \forall A \in G.
> > $$
> > ***
> >
> > # 7. Mean and variance calculation in GyroBN is flexible.
> >
> > Any method of calculating mean and variance can be used in our GyroBN, as the core goal of Riemannian batch normalization is to achieve normalization. Our GyroBN adopts the same mean calculation methods as other batch normalizations on Grassmannian and hyperbolic spaces.
> >
> > For the Grassmannian implementation, we follow previous work [f-h] and use the Karcher flow with a single step. In the hyperbolic implementation, we utilize Alg. 1 from [a] for fast computation, consistent with the RBN-H implementation. The efficiency gain of our method compared to other BN methods arises from the relatively fast computation of normalization defined by the gyro operations, as detailed in our common response **CQ#1**.
> > ***
> >
> > # References
> >
> > > [a] Differential geometry of Grassmann manifolds
> > >
> > > [b] Hyperbolic neural networks
> > >
> > > [d] Manifoldnorm: Extending normalizations on Riemannian manifolds
> > >
> > > [e] Differentiating through the Fréchet mean
> > >
> > > [f] A Lie group approach to Riemannian batch normalization
> > >
> > > [g] Riemannian batch normalization for SPD neural networks
> > >
> > > [h] SPD domain-specific batch normalization to crack interpretable unsupervised domain adaptation in EEG

---

> > > ### Comment · Reviewer_oCvr · 2024-11-22
> > > **Thanks**
> > >
> > > Thank you for response. The extra experiments on condition numbers and covariate shifts are very fascinating, and I believe would strengthen the paper. It could provide a better explanation of why batch norm is useful for geometric deep learning models. I hope that this will inspire future work to do similar analyses, which seem to be missing from many works in geometric deep learning. Overall I am still positive about the paper.

---

> ### Author Response · Authors · 2024-11-23
>
> Thanks for the instant reply and thoughtful reviews. We have added the discussion on covariate shift and condition number in App. F.7 and F. 8, respectively. These are included in the revised version of the paper. 😊

---

### Author Response · Authors · 2024-11-19
**Common Response**

We thank all the reviewers for their constructive suggestions and valuable feedback 👍.  Below, we address the common questions (CQs).

**Remark:** All references cited in the rebuttal correspond to the numbering in the revised manuscript.
***

# CQ#1: Our GyroBN could be more efficient than other RBN methods. (Reviewers $\textcolor{purple}{oCvr}$ and $\textcolor{green}{Yeux}$)

## GyroBN vs. other BN methods on the Grassmannian

As the Riemannian computations over the Grassmannian $\mathrm{Gr}(p,n)$ involve matrix decompositions, the disparity in efficiency is more evident. The following is our detailed discussion.

**Table A**: Average training time (seconds/epoch) for GyroGr with and without different Grassmannian BNs. The dimensions of the input Grassmannian matrices for the BN layer are also reported.

|       Methods       |  HDM05 ($47 \times 10$) |  NTU60 ($75 \times 10$)  |  NTU120 ($75 \times 10$) |
|:-------------------:|:------:|:-------:|:-------:|
|        GyroGr       |  2.19  |  50.92  |  80.72  |
| GyroGr-ManifoldNorm | 33.08  | 242.12  | 409.48  |
|      GyroGr-RBN     | 33.88  | 242.63  | 410.08  |
|    GyroGr-GyroBN    |  3.10  |  59.55  | 108.92  |

**Results of Training Efficiency:** Tab. A shows the training efficiency of GyroGr with and without different Grassmannian BNs. Compared to other Grassmannian BNs, GyroBN is significantly more efficient due to the simplicity of the gyro operation.

**Analysis:** The key difference between GyroBN (Alg. 1) and the ManifoldNorm [a, Algs. 1-2] or RBN [b, Alg. 2] lies in their methods for centering, biasing, and scaling. GyroBN uses gyro operations, while ManifoldNorm and RBN rely on Riemannian operators, such as parallel transport and the Riemannian logarithmic and exponential maps. This distinction underpins the efficiency of GyroBN. The three primary contributing factors, ranked by importance, are as follows:

1. **Riemann vs. Gyro:** The Riemannian operators over the Grassmannian involve computationally expensive processes like SVD decomposition or matrix inversion (see Tab. 7 for Riemannian exp and log, and [c, Thm. 2.4] for parallel transport). Consequently, ManifoldNorm and RBN require multiple SVD or matrix inversion operations. In contrast, GyroBN is relatively simpler. As discussed in Sec. 6.1, the gyro operation can be further simplified, and the involved SVD is performed on a reduced $p \times p$ matrix instead of $n \times p$. Additionally, as noted in F.1.2, computationally intensive matrix exponentiation is efficiently approximated using the Cayley map.

2. **Reduced Matrix Products:** As shown in Tab. 7, each Riemannian operator involves several matrix products over $n \times p$ matrices. GyroBN reduces these to matrix products over $(n-p) \times p$ or $p \times p$ matrices, which are computationally more efficient. See Prop. 6.1 for a detailed example.

3. **Optimization:** The biasing parameter optimization in GyroBN is simpler. ManifoldNorm uses an $n \times n$ orthogonal matrix for biasing, and RBN employs an $n \times p$ Grassmannian matrix, both requiring Riemannian optimization. In contrast, GyroBN applies trivialization tricks (parameterizing manifold data via the Riemannian exp at the identity [d], as shown in Eqs. (50-51)), making the biasing parameter a $(n-p) \times p$ Euclidean matrix. Furthermore, Eqs. (50-51) and Eq. (22) share a similar form, allowing them to be jointly simplified. Although RBN could adopt similar trivialization for biasing, this would introduce an additional Riemannian exp step, leaving little advantage over the Riemannian optimization. In summary, GyroBN benefits from joint simplification with trivialization, whereas the other two Grassmannian BN methods require additional Riemannian optimization.

## GyroBN vs. RBN [b] on the hyperbolic space

**Table B**: Average training time (seconds/epoch) for HNN with and without different hyperbolic BNs. The dimension of the input feature in the BN layer is 128.
|    Methods   |  Cora  | Disease | Airport | Pubmed |
|:------------:|:------:|:-------:|:-------:|:------:|
|      HNN     | 0.0323 |  0.0271 |  0.054  | 0.1253 |
|   HNN-RBN-H  | 0.0905 |  0.0883 |  0.1215 | 0.3416 |
| HNN-GyroBN-H | 0.0757 |  0.0842 |  0.119  | 0.3351 |

Recalling Tab. 7, the Riemannian logarithmic and exponential maps over the hyperbolic space contain gyro operations. Therefore, our GyroBN is expected to be more efficient than the RBN. Tab. B reports the average training time per epoch, indicating that our GyroBN is more efficient than RBN [b] on the hyperbolic space.

*Remark: The above discussion has been added to the revised submission (App. F. 5.)*
***

# References

> [a] Manifoldnorm: Extending normalizations on Riemannian manifolds.
>
> [b] Differentiating through the Fréchet mean.
>
> [c] The geometry of algorithms with orthogonality constraints.
>
> [d] Trivializations for gradient-based optimization on manifolds

---

### Meta-Review · Area_Chair_q5js · 2024-12-19

**Metareview:**

This study introduces a general Riemannian batch normalization framework on gyrogroups. Experiments demonstrate its effectiveness. After the response, it receives two borderline accept and two accept. Reviewers are satisfied with its novelty, presentation, and good results. I agree with them and think the current manuscript meets the requirements of this top conference. Please also incorporate the response in the revised paper.

**Additional Comments On Reviewer Discussion:**

The concerns are well addressed by the authors in the response. I suggest Accept.

---

### Decision · Program_Chairs · 2025-01-22

Accept (Poster)